# Fair Continuous Resource Allocation with Equality of Impact

**Blossom Metevier**[*]
University of Massachusetts
Amherst, MA 01002
blossom.metevier@gmail.com

**Dennis Wei**
IBM Research
San Jose, CA 95120
dwei@us.ibm.com

**Karthikeyan Natesan Ramamurthy**
IBM Research
Yorktown Heights, NY 10598
knatesa@us.ibm.com

**Philip S. Thomas**
University of Massachusetts
Amherst, MA 01002
pthomas@cs.umass.edu

## Abstract

Recent works have studied fair resource allocation in social settings, where fairness is judged by the *impact* of allocation decisions rather than more traditional minimum or maximum thresholds on the allocations themselves. Our work significantly adds to this literature by developing continuous resource allocation strategies that adhere to *equality of impact*, a generalization of equality of opportunity. We derive methods to maximize total welfare across groups subject to minimal violation of equality of impact, in settings where the outcomes of allocations are unknown but have a diminishing marginal effect. While focused on a two-group setting, our study addresses a broader class of welfare dynamics than explored in prior work. Our contributions are threefold. First, we introduce *Equality of Impact (EoI)*, a fairness criterion defined via group-level impact functions. Second, we design an online algorithm for non-noisy settings that leverages the problem's geometric structure and achieves constant cumulative fairness regret. Third, we extend this approach to noisy environments with a meta-algorithm and empirically demonstrate that our methods find fair allocations and perform competitively relative to representative baselines.

## 1 Introduction

Resource allocation problems are those in which a limited supply of some *resource* must be distributed across groups with varying welfare needs. In social applications, allocation decisions often have significant fairness and equity implications. For example, in urban planning, the efficient and equitable management of services like water and power distribution across urban and rural areas is critical [Turley, 2023, Prieto, 2021]. Similarly, in economic aid distribution, large-scale funds must be strategically allocated to different regions to maximize economic outcomes while reducing issues such as discrepancies in unemployment or poverty levels [Tang and Zhang, 2017].

Many frameworks have been developed to define fairness in resource allocation Singh [2020], Mashiat et al. [2022]. For example, Allocative Equality of Opportunity (AEoO) [Elzayn et al., 2019] adapts Equality of Opportunity [Hardt et al., 2016], a fairness notion from classification, to allocation problems. AEoO requires that qualified individuals from different groups have equal probabilities of receiving a resource. While this criterion promotes fair access to opportunity, it depends on the

---

[*]Correspondence should be directed to this author.

39th Conference on Neural Information Processing Systems (NeurIPS 2025).

ability to measure individual-level probabilities of success, which can be unavailable or infeasible to estimate in practice Corbett-Davies et al. [2023]. Additionally, many real-world settings naturally involve collective or systemic allocation impacts, where fairness must instead be evaluated at the group level.

For example, in water allocation problems studied by Crespo et al. [2022], social welfare is closely linked to ecological outcomes such as the preservation of environmental flows and overall ecosystem health. These outcomes, which we refer to as *impacts*, can be quantified using metrics like ecosystem health scores or improvements in water quality and pollutant reduction. Similar considerations arise in resource management, where fairness is often assessed through measures such as the relative deprivation index, which evaluates group-level deviations from equitable states [Zhang et al., 2022, Sullivan, 2002]. Together, these examples highlight the need for fairness notions that account for the collective consequences of allocation decisions on groups rather than individuals. To address this need, we introduce *Equality of Impact (EoI)*, a fairness notion that generalizes AEoO by shifting the focus from individual opportunity to the measurable impacts of allocations on groups.

We study continuous resource allocation settings that require EoI. Our goal is to create an algorithm, or a learning agent, able to make allocation decisions that maximize total welfare while ensuring EoI across two groups. We consider learning in an online censored-feedback setting where the agent makes allocation decisions over a series of independent rounds, and receives only feedback on the resources deployed to each group. Critically, the agent has no prior knowledge of which allocations violate fairness or maximize welfare, and only observes the resulting welfare and impact outcomes after each allocation. Therefore, to determine the best allocation, the agent must strategically explore sub-optimal allocations while minimizing the impact disparities across the groups. We assume that the relationship between allocations and outcomes conforms to a class of functions characterized by diminishing marginal returns (as more resources are allocated to a group, the incremental gain decreases). While this assumption informs the structure of the problem, the agent must still balance fairness and welfare objectives when exploring.

**Contributions** First, we propose Equality of Impact (EoI), a generalization of Allocative Equality of Opportunity that defines fairness through application-specific impact functions based on group-level outcomes (Section 2). Second, we design an online algorithm for the non-noisy setting that exploits the problem's geometric structure (Sections 3 and 4); we prove this algorithm achieves constant $O(1)$ cumulative fairness regret (Section 5). Finally, we propose a meta-algorithm for noisy settings (Section 6) and provide an empirical validation (Section 7). Our experiments confirm our theoretical results in the non-noisy setting, and show that our algorithm for noisy settings can compare favorably to standard baseline methods across a range of datasets.

## 2    Problem formulation

We consider a learning agent that, at each round $t \in (1, 2, \ldots, T)$, receives $q \in \mathbb{R}_{\geq 0}$ units of continuous resource. The agent distributes this resource across groups $A$ and $B$ according to allocation scheme $a^t = \langle a_A^t, a_B^t \rangle$, where $a_i^t$ represents the amount of resources allocated to group $i \in \{A, B\}$ at round $t$. When the round is clear from context, we drop the $t$ superscript, e.g., $a_A$. Allocations must be greater than zero and sum to no more than $q$ at each round, thus defining the action set $\mathcal{A}$ of allowable allocations $\mathcal{A} = \{a^t : a_A^t \geq 0, a_B^t \geq 0, \ a_A^t + a_B^t \leq q\}$.

Each group $i \in \{A, B\}$ has a *reward* function $r_i : [0, q] \to \mathbb{R}$, which quantifies the welfare derived from its allocation, and an *impact* function $h_i : [0, q] \to \mathbb{R}_{\geq 0}$, which quantifies a different implication of the allocation, such as a societal benefit or harm. We discuss impact functions in Section 2.1.

At each round, the agent makes an allocation decision and receives outcomes from the reward and impact functions. We assume the relationship between allocations and outcomes is characterized by diminishing marginal returns: As more resources are allocated to any function in this class, the incremental benefit gained does not increase. Formally,

**Definition 1** (Marginal Returns Class). A function $f : [0, q] \to \mathbb{R}$ belongs to the Marginal Returns Class if it satisfies the following three properties:

1.  Non-decreasing: $\forall \langle a_i, a_i' \rangle \in [0, q]^2, \langle a_i < a_i' \rangle \implies (f(a_i) \leq f(a_i'))$.
2.  Concavity: For all $\alpha \in [0, 1]$ and $\langle a_i, a_i' \rangle \in [0, q]^2, f\big(\alpha a_i + (1-\alpha)a_i'\big) \geq \alpha f(a_i) + (1-\alpha)f(a_i')$.

3. Continuity: $f$ is continuous with respect to $a \in [0, q]$.[2]

This modeling choice is common in economics, where increasing inputs typically yield progressively smaller gains due to factors like resource saturation or efficiency losses [Nicholson and Snyder, 1997]. It also reflects real-world phenomena observed in applications such as water management [Klocke et al., 2006] and bandwidth allocation [Shenker, 1995]. We also assume that the reward and impact functions are known when no resources are allocated.

**Assumption 1** (Reward and Impact Function Properties). For all $i \in \{A, B\}$, $r_i$ and $h_i$ belong to the Marginal Returns Class, $r_i(0)$ is known, and $h_i(0) = 0$.

We consider the utilitarian welfare function, or utility function for short, which is the sum of rewards for each group. We denote by $x^\star$ a *reward-maximizing* allocation that maximizes utilitarian welfare:

$$u(a^t) = r_A(a_A^t) + r_B(a_B^t), \quad x^\star \in \operatorname*{argmax}_{a \in \mathcal{A}} u(a). \tag{1}$$

## 2.1 Allocative fairness

Elzayn et al. [2019] defines an allocation to be unfair if among eligible individuals in two groups, those from one group have a higher probability of receiving a resource than those from the other group. They refer to this notion as Allocative Equality of Opportunity (AEoO). In the predictive policing example from their paper, AEoO implies that, conditioned on committing a crime (eligibility), the probability of apprehension should not depend on the district (group) where the crime occurs.

Our notion of *Equality of Impact* (EoI) broadens AEoO to incorporate metrics that assess group-level outcomes, and not just individual-level probabilities. We use the impact function $h_i$ introduced previously to measure the application-specific effect an allocation has on group $i$. We say an allocation scheme is fair if it satisfies approximate EoI across groups, defined as follows:

**Definition 2** (Equality of Impact). Define $A^G$ as:

$$A^G = \{a \in \mathcal{A} : |h_A(a_A) - h_B(a_B)| \leq G\}$$

for $G \in [0, \infty)$. Then allocation scheme $a$ satisfies EoI if $a \in A^G$.

Definition 2 captures the principle that it is unfair if one group consistently has better outcomes than those in another group. Notice that setting $G \geq \max(h_A(q), h_B(q))$ imposes no fairness constraint on the allocations, and setting $G = 0$ requires strict EoI. For intuition, below we provide two examples: Mapping AEoO to EoI, and framing a water allocation problem in the context of EoI.

**Example: Allocative Equality of Opportunity**   As stated previously, AEoO formalizes the intuition that it is unfair if eligible individuals from one group have a higher probability of receiving resources than individuals in another. Specifically, Elzayn et al. [2019] assume that each group $i$ contains $c_i$ *candidates*, where $c_i \sim C_i$ is a random variable indicating the number of individuals the agent would like to receive resources. Additionally, associated with each group $i$ is a *discovery model* $\mathrm{disc}(a_i, c_i)$ which, given $a_i$ resources and $c_i$ total candidates, returns the number of candidates that receive a resource. Given a group's allocation and discovery model, for all groups $i$ and $j$, Elzayn et al. [2019] define AEoO as

$$\left| f_i(a_i, \mathrm{disc}(\cdot), C_i) - f_j(a_j, \mathrm{disc}(\cdot), C_j) \right| \leq G,$$

where $C_i$ is a fixed but unknown marginal candidate distribution, and

$$f_i(a_i, \mathrm{disc}(\cdot), C_i) = \mathbf{E}_{c_i \sim C_i} \left[ \frac{\mathrm{disc}(a_i, c_i)}{c_i} \right]$$

is the expected probability that a random candidate from group $i$ receives a unit of resource at allocation $a_i$. Then AEoO can be mapped to EoI (Definition 2) by setting $h_i(a_i) = f_i(a_i, \mathrm{disc}(\cdot), C_i)$. Similarly, if the reward function is the expected number of discovered candidates $r_i(a_i) = \mathbf{E}_{c_i \sim C_i}[\mathrm{disc}(a_i, c_i)]$, the utilitarian welfare function (1) maps to theirs. Appendix A details how the impact and reward functions in Elzayn et al. [2019] belong to the Marginal Returns Class.

**Example: water allocation**   Water allocation problems often require balancing efficiency and fairness, particularly in drought-prone regions or transboundary water management, where unequal

---

[2]One might model problems with large numbers of discrete resource units (e.g., millions) as being continuous.

distribution can exacerbate shortages for vulnerable communities. These problems are commonly modeled with diminishing marginal returns because additional water tends to result in progressively smaller benefits [Klocke et al., 2006]. In water allocation, fairness can be measured by metrics like the relative deprivation index [Zhang et al., 2022, Sullivan, 2002], which quantifies an allocation's deviation from an ideal equitable distribution. We can define a logarithmic impact function $h_i(x) = \log(1 + \frac{a_i}{d_i})$ (concave, non-decreasing) to model the relative deprivation index of group $i \in \{A, B\}$, where $d_i$ is a predefined fair reference level (e.g., the minimum required water to meet basic needs).

## 2.2  Optimization objective

The agent's goal is to find an allocation $a^\star$ that maximizes utility subject to satisfying EoI:

$$a^\star \in \arg \max_{a \in A^G} u(a). \tag{2}$$

We refer to allocations that satisfy (2) as optimal allocations. Since the reward and impact functions are stationary (do not depend on round $t$), $a^\star$ satisfies (2) for all rounds $t$ if it does so at one round.

In our setting, both the reward and impact functions are unknown to the agent a priori (except for the value at zero allocation), and at each round the agent only receives single-point outcomes indicating the effectiveness of its allocation decisions. Because of this, the agent must reason about the best allocations to make at each round, incurring *regret*, i.e., the gap between the outcomes of its decisions and the outcomes that could have been achieved by allocating $a^\star$.

Our theoretical analysis in Section 5 considers *cumulative fairness regret*, which measures the total extent to which allocations deviate from EoI over $T$ rounds. To quantify this, let $h(a) = h_A(a_A) - h_B(a_B)$ be the difference in impact between groups, and define the *instantaneous fairness regret* as $g(a, G) = \max(0, |h(a)| - G)$, which is zero when an allocation satisfies EoI. The cumulative fairness regret is then $R_{\text{fair}}(T) = \sum_{t=1}^{T} g(a^t, G)$. When clear from context, we write $g(a, G)$ simply as $g(a)$.

## 3  Characterizing optimal allocations

In this section we argue that it suffices to search the set of allocation schemes utilizing all resources, denoted $\mathcal{Q} = \{a \in \mathcal{A} : a_A + a_B = q\}$, to find an optimal allocation. To do so, we first show that there always exists a (possibly singleton) set of allocation schemes in $\mathcal{Q}$ that satisfies EoI and contains an optimal allocation. We also show that this set corresponds to a single closed interval, a property our algorithm (described in Section 4.4) leverages to efficiently search for an optimal solution. Lastly, we consider the utility-maximizing allocation schemes that utilize all resources, denoted $X$. Our methods (detailed in Sections 4.4 and 6) rely on estimates of $X$ to efficiently find optimal allocations. All proofs for theorems and lemmas in this section are located in Appendix B.

Denote the set of fair allocation schemes that utilize all resources as

$$\mathcal{Q}^G = \{a \in [0, q] : \langle a, q - a \rangle \in \mathcal{Q} \cap A^G\}.$$

We first show that a fair allocation scheme utilizing all resources always exists. We then show that $\mathcal{Q}^G$ is a closed interval and always contains at least one optimal allocation.

**Lemma 1.** *Given Assumption 1, and for all $G \geq 0$, $\mathcal{Q}^G$ is non-empty.*

**Theorem 1.** *Given Assumption 1, $\mathcal{Q}^G$ is a single closed interval.*

Theorem 2 shows that there exists an optimal allocation in the set of fair allocations that utilize all resources. This implies that we can restrict our search to the set of allocation schemes that sum to $q$.

**Theorem 2.** *Given Assumption 1, there exists an $a^\star \in \mathcal{Q}^G$.*

Moving forward, we reference allocation schemes based on the allocation assigned to group $A$, i.e., allocation $x$ refers to the allocation scheme $\langle x, q - x \rangle$. Lastly, we define the set of utility-maximizing allocations that utilize all resources as

$$X = \underset{x \in [0, q]}{\arg\max} \, u(\langle x, q - x \rangle),$$

and show in Lemma 2 that $X$ is a closed interval and a subset of all utility-maximizing allocations.

**Lemma 2.** *Given Assumption 1, $X$ forms a single closed interval in $[0, q]$ and is a subset of $\arg\max_{x \in \mathcal{A}} u(x)$.*

# 4 Strategies for deterministic settings

In the noise-free setting, at each round $t$, the outcomes observed by the agent after sampling an allocation correspond to the exact values of the respective impact and reward functions. Therefore, bounds can be constructed that contain the impact and reward functions with absolute certainty. In this section, we derive these bounds (Section 4.1), use them to derive estimates of $X$ and $\mathcal{Q}^G$ (Section 4.3), and present an algorithm that leverages them to compute $a^\star$ (Section 4.4). All proofs for this section are located in Appendix D.

## 4.1 Properties of the marginal returns class

By Assumption 1, the reward and impact functions belong to the Marginal Returns Class. We show in Appendix C, Lemma 7, that properties of the chords connecting allocations can be used to derive strict bounds within which these functions must lie (see Figure 1 for an example). Following Lemma 7, let $L_{h_i}^t$, $U_{h_i}^t$ and $L_{r_i}^t$, $U_{r_i}^t$ denote the respective lower and upper bound functions of the impact and reward function for group $i \in \{A, B\}$ at round $t$. For example, if the black curve in Figure 1 is $h_A$, then after the allocation decisions $x^0$ and $x^1$, the upper bound function $U_{h_A}^{t=2}$ is represented by the orange line, and the lower bound function $L_{h_A}^{t=2}$ is represented by the blue line. Appendix C, Property 6, details additional characteristics of these bounds, including their continuity, monotonicity, and concavity, which are later used to construct estimates of $X$ and $\mathcal{Q}^G$.

## 4.2 Estimating the reward-maximizing set

At any round $t \in \{1, \ldots, T\}$, the upper and lower bounds of the utility function can be described as the summation of the upper and lower bounds of the reward functions, since these bounds are continuous and non-decreasing (Property 6):

$$L_u^t(x) = L_{r_A}^t(x) + L_{r_B}^t(q - x), \quad (3)$$
$$U_u^t(x) = U_{r_A}^t(x) + U_{r_B}^t(q - x).$$

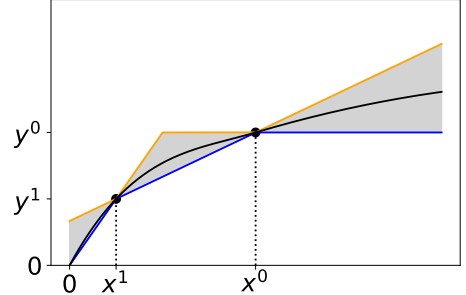

Figure 1: Example of a function belonging to the Marginal Returns Class (black curve) with two sampled outcomes $y_0, y_1$ at allocations $x_0, x_1$. The concavity and non-decreasing properties permit lower and upper bounds (the blue and orange lines, respectively) to be constructed, between which the non-sampled portions of the function must lie.

Our goal is to use the utility function bounds at round $t$ to construct an estimate of $X$, denoted $\hat{X}_t$, that contains all allocations that could be in $X$. To construct $\hat{X}_t$, denote the largest lower utility bound as $\ell_t = \max_{x \in [0,q]} L_u^t(x)$. Any allocation with an upper utility bound less than $\ell_t$ cannot be a utility-maximizing allocation, because there exists at least one allocation where the utility is guaranteed to be at least $\ell_t$. We therefore include only allocations with upper utility greater than $\ell_t$ in $\hat{X}_t$, and show in Lemma 3 that $\hat{X}_t$ is a single closed interval:

$$\hat{X}_t = \{x \in [0, q] : U_u^t(x) \geq \ell_t\}. \quad (4)$$

**Lemma 3.** *Given Assumption 1, for all $t > 1$, the set $\hat{X}_{t-1}$ is a single closed interval in $[0, q]$.*

## 4.3 Estimating the fair set

We construct two different estimates of $\mathcal{Q}^G$ at each round. The first estimate is a subset of $\mathcal{Q}^G$, called the *guaranteed fair set*, and contains all allocations in $\mathcal{Q}$ guaranteed to be fair at round $t$. The second estimate is a superset of $\mathcal{Q}^G$, denoted the *potentially fair set*, and contains all guaranteed fair allocations in $\mathcal{Q}$, as well as allocations that are potentially fair.

To define the guaranteed fair set, we establish a property that ensures fairness of an allocation:

**Property 1.** Given Assumption 1, if an allocation $x$ is contained in the set $\{x \in [0, q] : |L_{h_A}^t(x) - U_{h_B}^t(q - x)| \leq G\} \cap \{x \in [0, q] : |L_{h_B}^t(q - x) - U_{h_A}^t(x)| \leq G\}$ at round $t$, then it is fair.

**Definition 3** (Guaranteed Fair Set). The guaranteed fair set at round $t$, denoted $A_t^-$, is the set containing only the guaranteed fair allocations at round $t$, i.e., the allocations that satisfy Property 1.

| **Algorithm 1:** Non-Noisy | **Algorithm 2:** `nextAlloc` |
|---|---|
| **Input:** Budget $q$, Final round $T$ | **Input:** Potential fair set $A^+ = [p_L, p_R]$, |
| 1   $\hat{X}_1 = [0, q]$, $A_1^- = [\,]$, $A_1^+ = [0, q]$, $a_1 = q/2$ |         Guaranteed fair set $A^-$, Potential |
| 2   **for** $t = 1$ **to** $T$ **do** |         reward-max set $\hat{X} = [\hat{x}_L, \hat{x}_R]$ |
| 3      $R_A = r_A(a_t)$, $R_B = r_B(q - a_t)$ | 1   **if** $\hat{x}_R < p_L$ **then** |
| 4      $H_A = h_A(a_t)$, $H_B = h_B(q - a_t)$ | 2      **return** $p_L$ |
| 5      `updateBounds`$(R_A, R_B, H_A, H_B, a_t)$ | 3   **else if** $p_R < \hat{x}_L$ **then** |
| 6      `updateEstimates`$(L_{h_A}^t, U_{h_A}^t, L_{h_B}^t, U_{h_B}^t)$ | 4      **return** $p_R$ |
| 7      $\hat{X}_t = \{x \in [0, q] : U_u^t(x) \geq \max\limits_{x' \in [0,q]} L_u^t(x')\}$ | 5   **else** |
| 8      $a_{t+1} = $ `nextAlloc`$(A_t^+, A_t^-, \hat{X}_t)$ | 6      $\mathcal{F} = (A^+ \cap \hat{X}) \setminus A^-$ |
| **Output:** $a_T$ | 7      **return** $x \in \arg\max_{x' \in \mathcal{F}} U_u(x')$ |

In Lemma 4, we show that for any round $t$, the guaranteed fair set is a single interval (or empty).

**Lemma 4.** *For all $t > 1$, $A_t^-$ is a single interval in $[0, q]$.*

The potential fair set consists of all allocations that could satisfy EoI, i.e., those for which feasible values of $h_A(x)$ and $h_B(q - x)$ exist at round $t$. We show that this set also forms a single interval.

**Definition 4.** The potential fair set at round $t$, denoted $A_t^+$, is the set of allocations $x$ for which there exists a $y_A \in [L_{h_A}^t(x), U_{h_A}^t(x)]$ and $y_B \in [L_{h_B}^t(q - x), U_{h_B}^t(q - x)]$ such that $|y_A - y_B| \leq G$.

**Lemma 5.** *For all $t > 1$, $A_t^+$ is a single interval in $[0, q]$.*

### 4.4 Algorithm

Algorithms 1 and 2 outline our method for finding optimal allocations under non-noisy outcomes. In each round, after sampling the reward and impact functions (Algorithm 1, lines 3 and 4), their bounds are updated (line 5) along with the estimates of $\mathcal{Q}^G$ and $X$ (lines 6, 7). Algorithm 2 then uses these estimates to select the next allocation. Specifically, if $\hat{X}$ lies entirely to the left or right of $A^+$, an $a^\star$ must be on the boundary of $\mathcal{Q}^G$, and the algorithm selects the corresponding boundary point on $A^+$ (lines 1–4). Otherwise, it maximizes the upper bound of the utility function over $(A^+ \cap \hat{X}) \setminus A^-$. Appendix F.5 discusses the computational efficiency of updating these bounds and estimators.

## 5 Regret analysis

We show that Algorithm 1 achieves bounded cumulative fairness regret. In addition to Assumption 1, we assume that the impact functions are well-behaved at the boundaries of their domain:

**Assumption 2.** Assume that the one-sided derivatives of the impact functions at $0$ and $q$ are finite, specifically $\partial^+ h_i(0), \partial^- h_i(q) \in \mathbb{R}$, where $i \in \{A, B\}$.

**Theorem 3.** *Given Assumptions 1 and 2, Algorithm 1 achieves a cumulative fairness regret $\mathcal{R}_{fair}(T) = O(1)$.*

*Proof.* See Appendix E. $\square$

## 6 Strategies for noisy settings

In this section, we build on the insights from Section 3 to develop a meta-algorithm that identifies optimal allocations in settings where the sampled allocation outcomes are noisy. Importantly, even under noise, the set $\mathcal{Q}^G$ remains guaranteed to contain at least one optimal allocation. Similar to the deterministic setting, this allows us to restrict our search space to $\mathcal{Q}^G$ and thereby reduce complexity.

Assume that instead of observing the true outcomes for the reward and impact functions, the agent receives noisy outcomes of the form $\hat{h}_i(x) = h_i(x) + \epsilon_{h_i}$ and $\hat{r}_i(x) = r_i(x) + \epsilon_{r_i}$ where $i \in \{A, B\}$, and each $\epsilon_{h_i}$ and $\epsilon_{r_i}$ is drawn independently from a zero-mean normal distribution. This assumption

is common in settings where noise arises from the accumulation of independent factors, as suggested by the central limit theorem.

Because the agent now receives noisy outcomes when sampling the reward and impact functions, the bounds derived in Lemma 7 are no longer guaranteed to contain the true functions. This implies our previous method of constructing the potential and guaranteed fair sets $A^+$, $A^-$ is not suitable. In this section, we show how these sets can be estimated given impact function estimates in the form of confidence intervals (in the frequentist case) or credible intervals (in the Bayesian case). We assume the user provides confidence or credible bounds and updates them as noisy outcomes are observed. Similarly, we assume that the user maintains upper and lower bounds on the utility function, $U_u$ and $L_u$. Using these estimates, we develop a method to identify an optimal allocation $a^\star$.

**Assumption 3.** For all $f \in \{h_A, h_B, u\}$, the lower and upper bound functions $L_f$ and $U_f$ satisfy the following properties:

1. Each $f$ is continuous over $[0, q]$.
2. For a significance level $\delta_f$ and for each $x \in [0, q]$, the provided bounds $[L_f(x), U_f(x)]$ are constructed such that they contain the true function $f(x)$ with a high degree of confidence or credibility of at least $1 - \delta_f$. The specific interpretation of this probability (frequentist or Bayesian) depends on the underlying estimation method.
3. As more observations are collected, the bound shrinks, i.e., $\forall\, x \in [0, q]$ and any round $t > 1$,

$$U_f^t(x) - L_f^t(x) \leq U_f^{t-1}(x) - L_f^{t-1}(x).$$

For concreteness, in Section 6.1 we provide an example using credible intervals as provided by Gaussian processes.

The bounding functions are not assumed to be monotone, which allows greater flexibility in updating the impact and reward functions. As a consequence, we must carefully define our estimates of the potential and guaranteed fair sets to handle non-monotonicity. At each round $t$, let

$$D_1^t = \{x \in [0, q] : |L_{h_A}^t(x) - U_{h_B}^t(q - x)| \leq G\},$$
$$D_2^t = \{x \in [0, q] : |L_{h_B}^t(q - x) - U_{h_A}^t(x)| \leq G\}.$$

Then the estimates of the potential fair and guaranteed fair sets are given by the following intervals:

$$\hat{A}_t^+ = [a_L^+, a_R^+], \quad \hat{A}_t^- = [a_L^-, a_R^-],$$

where

$$a_L^+ = \min\left(D_1^t \cup D_2^t\right), \qquad\qquad a_R^+ = \max\left(D_1^t \cup D_2^t\right),$$
$$a_L^- = \min\left(D_1^t \cap D_2^t\right), \qquad\qquad a_R^- = \max\left(D_1^t \cap D_2^t\right).$$

These sets include all allocations that are potentially (or guaranteed) fair (with a high degree of confidence or credibility), in addition to some that are not.

Algorithm 3 describes our method. Similar to Algorithm 1, we first observe (now noisy) outcomes of the reward and impact functions. These are passed to a user-provided function that updates the bounds. We then construct estimates of the guaranteed and potential fair sets as discussed above, as well as the set of reward-maximizing allocations.

## 6.1 Gaussian Process Estimators

To illustrate Algorithm 3 for noisy settings, we consider using Gaussian Processes (GPs) to estimate the reward and impact functions because they allow a natural way to maintain credible bounds. Given noisy observations, a GP models function $f$ as $f(x) \sim \mathcal{GP}(m(x), k(x, x'))$, where $m$ is the mean function and $k$ is the kernel. The posterior distribution $\hat{f}(x) \sim \mathcal{N}(\mu_f(x), \sigma_f^2(x))$ provides mean and variance estimates that allow for the construction of credible bounds

$$L_f(x) = \mu_f(x) - \beta\sigma_f(x) \quad \text{and} \quad U_f(x) = \mu_f(x) + \beta\sigma_f(x),$$

where $\beta$ controls the width of the credible interval.

Notice that the GP estimators of the reward and impact functions satisfy Assumption 3. First, GPs with commonly used kernels, such as the radial basis function (RBF) and Matérn kernels, produce function

---

**Algorithm 3:** `Noisy`

**Input:** Continuous resource budget $q$, final round $T$, bound estimator function `update_bounds`

1   $\hat{X}_1 = [0, q], A_1^- = [\,], A_1^+ = [0, q], a_1 \in (0, q)$
2   **for** $t = 1$ **to** $T$ **do**
3      $\hat{R}_A = r_A(a_t) + \epsilon_{r_A}, \hat{R}_B = r_B(q - a_t) + \epsilon_{r_B}$
4      $\hat{H}_A = h_A(a_t) + \epsilon_{h_A}, \hat{H}_B = h_B(q - a_t) + \epsilon_{h_B}$
5      $\{L_f^t, U_f^t\} = $ `update_bounds`$(\hat{R}_A, \hat{R}_B, \hat{H}_A, \hat{H}_B, a_t)$
6      $\hat{A}_t^+ = [\min(D_1^t \cup D_2^t), \max(D_1^t \cup D_2^t)], \quad \hat{A}_t^- = [\min(D_1^t \cap D_2^t), \max(D_1^t \cap D_2^t)]$
7      $\mathcal{X} = \{x \in \hat{X}_{t-1} : U_u^t(x) \geq \max_{x' \in \hat{X}_{t-1}} L_u^t(x')\}, \quad \hat{X}_t = [\min \mathcal{X}, \max \mathcal{X}]$
8      $\mathcal{F} = \hat{A}_t^+$ **if** $\hat{X}_t \cap \hat{A}_t^+ = \emptyset$ **else** $\hat{X}_t \cap \hat{A}_t^+$
9      $a_t = \operatorname{argmax}_{x \in \mathcal{F}} U_u^t(x)$
10  **return** $a_T$

---

estimates that are (probabilistically) smooth and continuous over $[0, q]$. Second, under Gaussian assumptions, the posterior distribution $\hat{f}(x) \sim \mathcal{N}(\mu_f(x), \sigma_f^2(x))$ implies that the probability of the true function value lying within the interval $[\mu_f(x) - \beta\sigma_f(x), \mu_f(x) + \beta\sigma_f(x)]$ is given by $1 - \delta_f$, where $\beta = \Phi^{-1}(1 - \delta_f/2)$ and $\Phi^{-1}$ is the inverse CDF of the standard normal distribution. This construction ensures that the credible interval $\Pr(f(x) \in [L_f(x), U_f(x)]) \geq 1 - \delta_f$. Finally, as more data points are sampled, the posterior variance of the GP does not increase, thereby satisfying the condition that the width of the credible bounds does not grow over time.

## 7 Empirical analysis

We test our agent's performance in various deterministic and noisy settings. We consider two research questions: **Q1**: Do the theoretical guarantees established in the noiseless setting hold empirically? **Q2**: In the presence of noise, can our method identify high-performing allocations effectively? Further experiment details can be found in Appendix F, including experiments related to reward-based regret.

**Environments**   We consider three instantiations of reward and impact functions, which we refer to as *environments*. In the *imbalanced-rewards* environment (IRE), we consider a scenario where one group's reward function consistently yields lower rewards at the same allocation than the other, making the reward-maximizing solution favor the higher-yielding group. To promote more equitable reward distribution, we set $r_A = h_A$ and $r_B = h_B$. In the *imbalanced-impact* environment (IIE), we assess how the algorithm manages similar reward functions that have imbalanced social impacts across groups. For example, two community policing initiatives with similar funding may achieve comparable crime reduction (reward), but one community may experience greater negative impact if increased policing heightens tensions and distrust between law enforcement and marginalized groups. Lastly, motivated by works that develop models showing the effects of water allocation under diminishing marginal returns [Klocke et al., 2006], we consider an environment in which reward could represent the crop yield per unit of land, and impact as the log-based formulation of the relative deprivation index. We term this the water allocation environment (WAE).

**Baselines**   For **Q1**, we aim to assess whether Algorithm 1 adheres to the theoretical guarantees established in Theorem 3. Because the objective is to verify consistency with this bound, baseline comparisons are not necessary; however, we additionally include results for two simple reference allocators:

- `Explore-then-Commit` (ETC): Assumes knowledge of Theorem 2 (there exists an optimal allocation using all resources), but not of Theorem 1 ($\mathcal{Q}^G$ is a closed interval). It operates in two phases. In the *explore* phase, the allocator performs a stratified random search: it divides $[0, q]$ into a fixed number of bins, samples one allocation from each, and evaluates their fairness and welfare. In the *commit* phase, it selects the allocation with the highest welfare among those that are maximally fair and uses it for all remaining rounds.

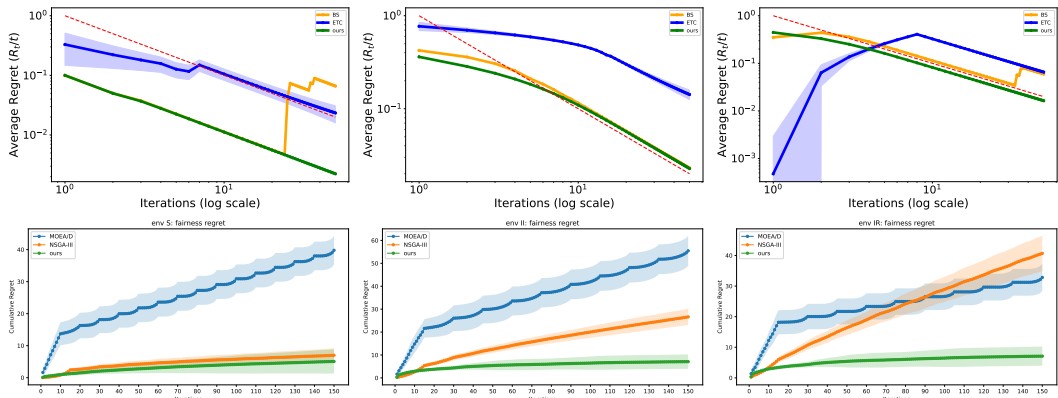

Figure 2: From left to right, environments WAE, IIE, and IRE are displayed, comparing our algorithm (green) to baselines. **Upper plots:** Average fairness regret ($R_t/t$) on a log-log scale over 50 rounds. The dashed red line represents the theoretical regret bound from Theorem 3. The stochastic ETC baseline is run 50 times (shaded area is standard deviation). Algorithm 1 and RS are deterministic and shown for a single trial. Our method empirically achieves the lowest regret and supports the theoretical bound. **Lower plots:** Cumulative fairness regret in the noisy setting, with shaded regions indicating the standard deviation across 50 trials.

- `Brent-Search` (BS): Has knowledge of Lemma 8 (that $h(x) = |h_A(x) - h_B(q - x)|$ is non-increasing on $[0, a_L]$ and non-decreasing on $[a_R, q]$), as well as Theorems 2 and 1. It simulates an oracle that first solves for $\mathcal{Q}^G$ and then finds the reward-maximizing allocation $a^\star$ within that set. It first uses a numerical root-finding routine to determine the boundaries of $\mathcal{Q}^G$, then applies Brent's optimization method to identify the welfare-maximizing allocation. BS commits to the final optimal allocation for all remaining rounds once the search completes.

For **Q2**, we compare Algorithm 3 in the noisy setting to Non-dominated Sorting Genetic Algorithm III (NSGA-III) Deb and Jain [2013] and Multi-Objective Evolutionary Algorithm Based on Decomposition (MOEA/D) Zhang and Li [2007], two common methods used in real-world continuous resource allocation problems requiring multiple objectives [Tang and Zhang, 2017, Miriam et al., 2021, Peng et al., 2022].

**Results and discussion** Figure 2 shows our results. In the noiseless setting (top row), the average fairness regret curves for our method align closely with the $O(1/T)$ reference lines in each environment, supporting our claim that the bound from Theorem 3 is asymptotically valid. Algorithm 1 empirically outperforms both baselines across all three environments. BS also follows the $O(1/T)$ trend but incurs a consistently higher regret than our approach and appears to suffer from numerical instability in the WAE environment (left), as shown by its sharp regret spike. The ETC baseline (blue) performs worst, illustrating the cost of its mandatory exploration phase. In the IRE environment (right), its regret increases significantly before it can "commit" to a good allocation, causing its average regret to converge much more slowly. In the noisy setting (bottom row), all algorithms were allocated a 150-query budget, representing 150 allocation decisions and their associated observed rewards and impacts. For a fair comparison, we tuned the number of generations and population sizes for the evolutionary baselines (MOEA/D, NSGA-III) to match this query budget. This ensured that each algorithm received the same total information about the environment. Across all environments, our method consistently accumulates less cumulative fairness regret than the baselines, with narrower confidence bands in IIE and IRE, indicating more stable performance. However, in WAE, the standard deviation intervals overlap, suggesting more comparable performance in this setting.

# 8   Additional related work

As stated previously, our work extends Allocative Equality of Opportunity (AEoO) [Elzayn et al., 2019] to include collective impacts of allocations, as defined by application-specific impact functions. This aligns with calls for context-aware fairness metrics that reflect real-world outcomes [Selbst et al., 2019]. In fair division, resources are often classified based on their characteristics: divisible (continu-

ous) or indivisible (discrete), and homogeneous (where only the quantity matters) or heterogeneous (where items have distinct identities) [Brandt et al., 2016, Brams and Taylor, 1996]. Our setting, which considers a continuous quantity of a single type of resource, falls in the divisible, homogeneous category. Classical solutions in fair division often assume known preferences or utility functions, and consider fairness notions such as proportionality, envy-freeness, or equitability [Moulin, 2004].

Our problem is an online learning task with censored feedback: the agent sequentially allocates resources without knowing the true reward and impact functions and observes only the outcomes corresponding to the chosen allocation. This setting aligns with frameworks such as multi-armed bandits (MAB) and online convex optimization (OCO) [Lattimore and Szepesvári, 2020, Agarwal et al., 2II, Shalev-Shwartz et al., 2012, Joseph et al., 2016]. Prior work on fair online resource allocation typically focuses on discrete allocations [Hossain et al., 2021]—we instead consider outcome-based fairness in a continuous allocation setting. This situates our problem within continuous MABs or bandit convex optimization, particularly for concave objectives [Hazan et al., 2007]. A related OCO approach by Yu and Neely [2020] studies convex objectives under convex constraints and achieves comparable constraint violation rates to Theorem 3. Equality of impact can be cast in their framework by taking the negatives of the concave reward and impact functions. However, the online learning problem studied in this work differs in the type of allocation feedback assumed: while Yu and Neely [2020] assumes gradient access, we derive bounds from point-wise evaluations of the reward and impact functions, without gradients.

The non-noisy setting is similar to non-stochastic bandit and online learning frameworks, where agents observe noise-free feedback. In these settings, sublinear regret can be achieved without algorithms designed for noise robustness [Cesa-Bianchi and Lugosi, 2006], and prior work demonstrates that exact feedback enables tight regret bounds [Yan et al., 2023, Zhao et al., 2023]. Our work adds to this literature by considering constraint violation (Theorem 3) in the non-noisy case.

## 9    Limitations and future work

Our analysis relies on simplifying assumptions that suggest directions for future work. First, our regret analysis only considers fairness regret, and assumes access to ground-truth feedback from reward and impact functions; extending it to stochastic environments and reward-based regret is a natural next step. In particular, we note that stronger assumptions on the reward functions, such as strict concavity [Hazan et al., 2007], smoothness conditions [Xu and Wang, 2021], or additional gradient information [Yu and Neely, 2020], could be used to achieve different reward-based regret rates. Second, we assume allocations affect only the corresponding group's welfare and impact. In reality, allocations often produce spillover or network effects. For example, improving one road can reduce congestion on connected roads and benefit multiple groups—capturing such interdependencies would require more complex tools such as game theory or network flow algorithms [Jackson and Zenou, 2015]. Finally, while we focus on a two-group setting relevant to inequity between disadvantaged and majority populations, extending our framework for learning allocations that satisfy Equality of Impact to multiple groups is an important direction for future work. Some of our current results already carry over: if the reward and impact functions satisfy Assumption 1, the fair set remains closed and convex, and welfare remains a sum of concave functions. The main challenge in higher dimensions lies in scaling the estimation of the fair sets and the utility-maximizing set.

## 10    Conclusion

We introduced Equality of Impact, a novel fairness criterion for resource allocation problems that extends equality of opportunity to group-level outcomes. We studied EoI in continuous allocation settings with diminishing marginal returns and developed Algorithm 1 for non-noisy settings that exploits this geometric structure. We show that Algorithm 1 achieves bounded cumulative fairness regret and extended our approach with a meta-algorithm for noisy settings. Empirical results support our theory, and demonstrate that in stochastic environments, our method performs competitively relative to strong baselines. Overall, this work provides a practical, provably effective framework for allocation problems where group-level impacts are the primary concern.

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

# A  Mapping Allocative Equality of Opportunity to Equality of Impact

Elzayn et al. [2019] consider two types of discovery models, i.e., models of how resources deployed to a particular group reach the candidates. These are termed the *random discovery* and *precision discovery* model. In the random discovery model, regardless of the number of units allocated to a given group, all individuals within that group are equally likely to be assigned a unit, regardless of whether they are a candidate:

$$\text{disc}(a_i, c_i) = a_i \frac{\mathbf{E}[c_i]}{m_i},$$

where $m_i$ are the total number of individuals in group $i$. In the precision discovery model, units of resource are given to actual candidates within a group, as long as there is sufficient supply of the resource:

$$\text{disc}(a_i, c_i) = \min(c_i, a_i).$$

Similar to our setting, Elzayn et al. [2019] consider utilitarian welfare, which in their setting can be described as the expected number of discovered candidates over all feasible allocations:

$$u(a, \text{disc}(\cdot), C) = \sum_{i \in \{A,B\}} \mathbf{E}_{c_i \sim C_i}[\text{disc}(a_i, c_i)].$$

The reward and impact functions defined using these discovery models are concave, continuous, and non-decreasing functions of the allocations, and therefore belong to the Marginal Returns class.

# B  Proofs for Section 3

Many of the proofs in this section are structured around a sequence of intermediate results, denoted [H1], [H2], ..., or [C1], [C2], ..., each of which establishes a key step in the argument. These claims are referenced by label throughout the proofs.

We begin this section by defining relevant properties used in later proofs. Consider the function $h_q : [0, q] \to \mathbb{R}$ defined as

$$h_q(a_A) = h_A(a_A) - h_B(q - a_A),$$

where $\langle a_A, a_B \rangle$ is an allocation scheme. Property 2 and 3 show that this function is continuous and non-decreasing over $[0, q]$.

**Property 2.** Given Assumption 1, the function $h_q$ is continuous.

*Proof.* This function is continuous over $[0, q]$ because the impact functions are continuous due to Assumption 1, and because subtraction preserves continuity. □

**Property 3.** Given Assumption 1, $h_q$ is non-decreasing.

*Proof.* Consider any $0 \le x_1 \le x_2 \le q$. We must show that $h_q(x_2) \ge h_q(x_1)$. Observe that

$$h_q(x_2) - h_q(x_1) = [h_A(x_2) - h_B(q - x_2)] - [h_A(x_1) - h_B(q - x_1)]$$
$$= [h_A(x_2) - h_A(x_1)] - [h_B(q - x_2) - h_B(q - x_1)].$$

Since $h_A$ is non-decreasing (by Assumption 1) and $x_2 \ge x_1$, we have that $h_A(x_2) - h_A(x_1) \ge 0$. Also, because $h_B$ is non-decreasing and $q - x_2 \le q - x_1$, it follows that

$$h_B(q - x_2) \le h_B(q - x_1) \implies h_B(q - x_2) - h_B(q - x_1) \le 0.$$

Therefore, $h_q(x_2) - h_q(x_1) \ge 0$, and $h_q$ is non-decreasing on $[0, q]$. □

We now prove Lemma 1, which states that the set of fair allocations schemes that sums to $q$, i.e., $\mathcal{Q}^G$ is non-empty.

**Lemma 1:** Given Assumption 1, and for all $G \ge 0$, $\mathcal{Q}^G$ is non-empty.

*Proof.* Notice that for any $G \geq 0$, $\mathcal{Q}^{G=0}$ is a subset of $\mathcal{Q}^G$. Therefore it suffices to show that $\mathcal{Q}^{G=0} \neq \emptyset$ to establish our claim. We do this by employing the Intermediate Value Theorem to show that there exists an allocation $x$ such that $h_q(x) = 0$, i.e., that $\mathcal{Q}^{G=0}$ contains at least $x$.

Consider the allocations $a^B = \langle 0, q \rangle$, where all resources are allocated to group $B$, and $a^A = \langle q, 0 \rangle$, where all resources are allocated to group $A$. Evaluating $h_q$ at these points, we get:

$$h_q(a_A^B) = h_A(0) - h_B(q) = 0 - h_B(q) \leq 0$$
$$h_q(a_A^A) = h_A(q) - h_B(0) = h_A(q) - 0 \geq 0.$$

This simplification relies on Assumption 1, specifically that the impact functions are non-decreasing and that $h_A(0) = 0$ and $h_B(0) = 0$. Because $h_q$ is continuous over $[0, q]$ (Property 2), the Intermediate Value Theorem guarantees there exists some allocation scheme $a$ such that

$$h_q(a_A) = h_A(a_A) - h_B(q - a_A) = 0.$$

Since $h_q(a_A) = 0$, it follows that $\forall\, G \geq 0, |h(a_A)| = 0 \leq G$. Therefore, $a \in \mathcal{Q} \cap A^{G=0}$, and for any $G \geq 0$, $\mathcal{Q}^G$ is non-empty. $\square$

Next, we provide a property of continuous monotone functions that will be used to show that $\mathcal{Q}^G$ is a closed interval (Theorem 1).

**Lemma 6.** *Let $f : [0, q] \rightarrow \mathbb{R}$ be a continuous, monotone function (either non-decreasing or non-increasing), and let $G \geq 0$. Then the set*

$$S = \{x \in [0, q] : |f(x)| \leq G\}$$

*is a (possibly degenerate or empty) closed interval in $[0, q]$.*

*Proof.* At a high level, we show that $S$ is an interval (steps [H1] and [H2]), and that $S$ is closed (step [H3]). To begin, we first re-write $S$ using the fact that $|f(x)| \leq G$ implies $-G \leq f(x) \leq G$:

$$S = \underbrace{\{x \in [0, q] : f(x) \leq G\}}_{S_1} \cap \underbrace{\{x \in [0, q] : f(x) \geq -G\}}_{S_2}.$$

[H1] **Monotonicity implies each set is (at most) one interval.** Assume without loss of generality that $f$ is continuous and *non-decreasing* (the argument for a non-increasing $f$ is similar). Then:

- Consider $S_1$. We show that it is either empty or can be represented as $[x', x_{\max}]$.
  - If $f(0) > G$, then for all $x \geq 0$, $f(x) > G$. Therefore $S_1$ is empty.
  - If $f(0) \leq G$, then $S_1$ is non-empty. Define $x_{\max} = \max S_1$. Because $f$ is non-decreasing, all values in $[0, x_{\max}]$ are therefore also contained in $S_1$.
- Consider $S_2$. Using a similar monotonicity argument, $S_2$ can only be empty (if $f(q) < -G$) or can be represented as $[x_{\min}, q]$, where $x_{\min} = \min S_2$.

[H2] **Intersection of intervals is an interval.** From [H1], we have that $S_1$ and $S_2$ are each intervals in $[0, q]$. The intersection of two intervals on the real line is itself an interval (possibly empty or a single point). Therefore $S = S_1 \cap S_2$ is at most one interval in $[0, q]$.

[H3] **Continuity ensures closedness.** Because $f$ is continuous (and monotone), $S_1$ and $S_2$ are closed in $[0, q]$:

- If $S_1$ is not empty, let $\{x_n\} \subseteq S_1$ be a sequence with $x_n \rightarrow x^* \in [0, q]$. By definition of $S_1$, $f(x_n) \leq G$ for all $n$. The continuity of $f$ implies that

$$f(x^*) = \lim_{n \to \infty} f(x_n) \leq G,$$

so $x^* \in S_1$ and therefore, $S_1$ is closed.
- A similar argument shows $S_2$ is closed.

Therefore their intersection $S = S_1 \cap S_2$ is also closed in $[0, q]$.

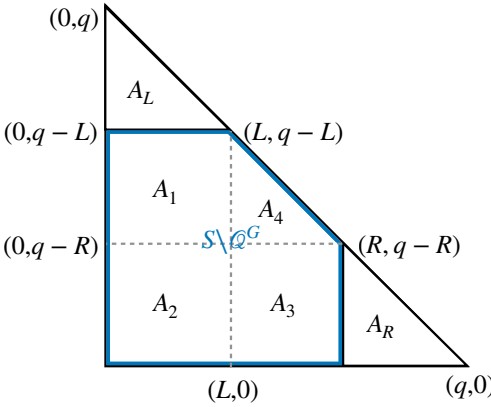

Figure 3: The triangular region above shows $\mathcal{A}$ and information relevant to the proof of Theorem 2. Possible allocations to group $A$ are on the $x$-axis and possible allocations to group $B$ are on the $y$-axis. The line connecting points $(L, q - L)$ and $(R, q - R)$ is the closed interval containing $\mathcal{Q}^G$. The regions $A_L$ and $A_R$ do not contain $A^G$. Lastly, $A_1, A_2, A_3,$ and $A_4$ are a partition of $S \setminus \mathcal{Q}^G$.

$\square$

Next, we prove Theorems 1 and 2.

> **Theorem 1:** Given Assumption 1, $\mathcal{Q}^G$, is a closed interval in $[0, q]$.

*Proof.* This follows directly from the above properties and Lemma 6. By Properties 2 and 3 (which require Assumption 1), $h_q$ is continuous and non-decreasing (on $[0, q]$). Then, by Lemma 6, the set $\{x \in [0, q] : |h_q(x)| \leq G\}$ is a closed interval in $[0, q]$. This is the definition of $\mathcal{Q}^G$.

$\square$

> **Theorem 2** Given Assumption 1, there exists an $a^\star \in \mathcal{Q}^G$.

*Proof.* To show our result, we first define a set $S$ such that $A^G \subseteq S$, where $A^G$ is the set of allocation schemes that satisfy EoI. We then demonstrate that for any allocation $a \in S \setminus \mathcal{Q}^G$, there exists an allocation $a' \in \mathcal{Q}^G$ such that $u(a') \geq u(a)$—consequently, a utility-maximizing allocation in $A^G$ exists in $\mathcal{Q}^G$, i.e., there exists an $a^\star \in \mathcal{Q}^G$. Throughout our proof we will refer to the diagram in Figure 3, which shows the feasible set of allocation schemes $\mathcal{A}$, and other relevant variables.

[H1] **Define a superset of $A^G$.** From Theorem 1 and Lemma 1, the set $\mathcal{Q}^G$ can be expressed as a non-empty interval $[L, R]$. Define the set $S$ as:

$$S = \{\langle a_A, a_B \rangle \in \mathcal{A} : a_A \leq R, \ a_B \leq q - L\}.$$

We claim that $A^G \subseteq S$. To show this, we define two disjoint subsets $A_L$ and $A_R$ whose union contains all points in $\mathcal{A} \setminus S$, and show that no point in $A_L \cup A_R$ belongs to $A^G$.

- *Defining disjoint sets of $S$.* Define:

$$A_L = \{\langle a_A, a_B \rangle \in \mathcal{A} : a_A < L, \ a_B > q - L\},$$
$$A_R = \{\langle a_A, a_B \rangle \in \mathcal{A} : a_A > R, \ a_B < q - R\}.$$

Since $S$, $A_L$, and $A_R$ are a partition of $\mathcal{A}$ (see Figure 3 for intuition), any allocation not in $S$ must be in $A_L \cup A_R$.

- *No allocation in $A_L$ or $A_R$ satisfies EoI*, i.e., $A^G \cap (A_L \cup A_R) = \emptyset$. For contradiction, assume that there exists an $a^L$ such that $a^L \in A_L$ and $a^L \in A^G$. Since $a^L \in A_L$, it follows that $a^L_A \in [0, L)$ and $a^L_B \in (q - L, q]$. Given that the impact functions $h_A$ and $h_B$ are non-decreasing (see Assumption 1), any allocation $a^L$ with $a^L_A \in [0, L)$ and $a^L_B \in (q - L, q]$ would result in $|h(a^L_A)| > G$, which contradicts the assumption that $|h(a^L)| \leq G$. Thus, if $a^L \in A_L$ then $a^L \notin A^G$, i.e., $A_L \cap A^G = \emptyset$. A similar argument holds for any $a^R \in A_R$. Consequently, if an allocation $a$ belongs to $A_L \cup A_R$, it cannot be in $A^G$. Therefore, $A^G$ must reside within $S$, because $A^G$ is non-empty (by Assumption 1) and because $S$, $A_L$, and $A_R$ together partition $\mathcal{A}$.

[H2] **For any allocation scheme $a \in S \setminus \mathcal{Q}^G$, there exists an allocation $a' \in \mathcal{Q}^G$ such that** $u(a') \geq u(a)$**.** Notice that we can write $S \setminus \mathcal{Q}^G$ as

$$S \setminus \mathcal{Q}^G = \{a \in S : a_A + a_B < q\}.$$

Define the following partition of $S$ (see Figure 3 for reference):

$$A_1 = \{a \in S : a_A \leq L, a_B \geq q - R\}$$
$$A_2 = \{a \in S : a_A < L, a_B < q - R\}$$
$$A_3 = \{a \in S : a_A \geq L, a_B \leq q - R\}$$
$$A_4 = \{a \in S : a_A > L, a_B > q - R\}$$

For any $a \in S \setminus \mathcal{Q}^G$, denote the unused units of resource as $s(a) = q - (a_A + a_B)$. Below, we show one way to reallocate this surplus to transform $a$ into an allocation $a' \in \mathcal{Q}^G$:

$$\begin{cases} a \in A_1 \cup A_4 & a' = (a_A + s(a), a_B) \\ a \in A_2 & a' = (R, a_B + s(a) - (R - a_A)) \\ a \in A_3 & a' = (a_A, a_B + s(a)). \end{cases}$$

In each case, $a'_A + a'_B = q$ and $L \leq a'_A \leq R$, ensuring $a' \in \mathcal{Q}^G$.

We now verify that $u(a) \leq u(a')$. For cases $a \in A_1 \cup A_4$ and $a \in A_3$, resources are added to either $a_A$ or $a_B$, and the reward functions being non-decreasing (Assumption 1) ensures that $u(a) \leq u(a')$. For case $a \in A_2$, we show that no resources are subtracted from $a_A$ or $a_B$, and therefore $u(a) \leq u(a')$.

- *Claim:* $a_A \leq a'_A$. For $a \in A_2$, we have that $a_A < L \leq R = a'_A$, where $a_A < L$ by definition of $A_2$, and $R = a'_A$ by the transformation function. Therefore a non-negative value is added to $a_A$.
- *Claim:* $a_B \leq a'_B$. For contradiction, assume that $a_B > a'_B$. Then the following inequalities must hold:

$$a_B > a'_B \quad \text{[our assumption for contradiction]}$$
$$a_B > a_B + s(a) - (R - a_A) \quad \text{[defn of } a'_B]$$
$$a_B > a_B + q - (a_A + a_B) - (R - a_A) \quad \text{[defn of } s(a)]$$
$$a_B > q - R. \quad \text{[simplify and rearrange]}$$

But by definition of $A_2$, $a_B < q - R$. This is a contradiction, and therefore, $a_B \leq a'_B$.

Therefore, non-negative resources are added to each element $a \in A_2$, and because the reward functions are non-decreasing, we also have $u(a) \leq u(a')$ for this case.

[H3] **An optimal allocation exists in $\mathcal{Q}^G$.** Define allocation schemes $b$ and $c$ that achieve maximum utility in $\mathcal{Q}^G$ and $S \setminus \mathcal{Q}^G$, respectively:

$$b \in \arg\max_{a \in \mathcal{Q}^G} u(a), \quad c \in \arg\max_{a \in S \setminus \mathcal{Q}^G} u(a).$$

[H2] implies that there exists an $a' \in \mathcal{Q}^G$ that achieves at least as much utility as $c$, i.e., $u(a') \geq u(c)$. Since $b$ maximizes $u(a)$ over $\mathcal{Q}^G$, we have:

$$u(b) \geq u(a') \geq u(c).$$

Therefore, the maximum utility over $A^G$ is achieved at $a^\star = b \in \mathcal{Q}^G$. Thus,

$$a^\star \in \arg\max_{a \in A^G} u(a) \in \mathcal{Q}^G.$$

□

Lastly, we show that Lemma 2, which will be used to justify Algorithm 1.

> **Lemma 2:** Given Assumption 1, the set $X$ satisfies the following properties:
> [C1] Let $X' = \{\langle x, q - x \rangle : x \in X\}$. Then $X' \subseteq \operatorname{argmax}_{x \in \mathcal{A}} u(x)$.
> [C2] $X$ forms a closed interval in $[0, q]$.

*Proof.* Define the following notation (some of which has been previously defined), which will be used throughout this proof:

$$X = \operatorname*{argmax}_{x \in [0,q]} u(\langle x, q - x \rangle), \quad X' = \{\langle x, q - x \rangle : x \in X\} = \operatorname*{argmax}_{x \in \mathcal{Q}} u(x),$$

$$X^\star = \operatorname*{argmax}_{x \in \mathcal{A}} u(\langle x, q - x \rangle), \quad \mathcal{Q} = \{x \in \mathcal{A} : x_A + x_B = q\}.$$

[C1] **Claim:** $X' \subseteq X^\star$. We prove the claim by first establishing that all allocations in $X'$ achieve the same utility as all allocations in $X^\star$. Then we leverage the relationship between $\mathcal{Q}$ and $\mathcal{A}$ to show that every allocation in $X'$ must also belong to $X^\star$.

> [H1] **Claim:** $\max_{x \in X^\star} u(x) = \max_{x \in X'} u(x)$. We prove the claim by constructing an allocation in $X'$ using an allocation in $X^\star$ that does not utilize all resources. Specifically, we show that such an allocation still belongs to $X^\star$, and because it utilizes all resources, it also belongs to $X'$, ensuring our claim.
>
> Choose any $x^\star \in X^\star$ such that $x_A^\star + x_B^\star < q$. By construction, $x^\star \notin \mathcal{Q}$ and thus $x^\star \notin X'$. Let the unused resources for $x^\star$ be defined as $s(x^\star) = q - (x_A^\star + x_B^\star)$. Construct a new allocation $x'$ as follows:
>
> $$x' = \langle x_A^\star + s(x^\star), x_B^\star \rangle.$$
>
> The utilities of these allocations are:
>
> $$u(x') = r_A(x_A^\star + s(x^\star)) + r_B(x_B^\star)$$
> $$u(x^\star) = r_A(x_A^\star) + r_B(x_B^\star).$$
>
> To prove $u(x') \geq u(x^\star)$, assume for contradiction that $u(x') < u(x^\star)$. Then:
>
> $$u(x') < u(x^\star) \quad \text{[assumption]}$$
> $$r_A(x_A^\star + s(x^\star)) + r_B(x_B^\star) < r_A(x_A^\star) + r_B(x_B^\star) \quad \text{[definition of utility]}$$
> $$r_A(x_A^\star + s(x^\star)) < r_A(x_A^\star) \quad \text{[simplify]}.$$
>
> By definition, $s(x^\star) \geq 0$, so $x_A^\star + s(x^\star) \geq x_A^\star$. From Assumption 1, the reward functions are non-decreasing, i.e., for any $x_1 \leq x_2$, $r_A(x_1) \leq r_A(x_2)$. This creates a contradiction, as $r_A(x_A^\star + s(x^\star)) \geq r_A(x_A^\star)$.
>
> Therefore, $u(x') \geq u(x^\star)$, which implies $x'$ is also contained in $X^\star$. Since all allocations in $X'$ have the same utility by definition, and all allocations in $X^\star$ have the same utility by definition, it follows that $\max_{x \in X^\star} u(x) = \max_{x \in X'} u(x)$.
>
> [H2] **Claim:** $X' \subseteq X^\star$. By [H1], all $x' \in X'$ achieve the same utility as all $x^\star \in X^\star$. Since $X' \subseteq \mathcal{Q}$, and $\mathcal{Q} \subseteq \mathcal{A}$, it follows that all $x' \in X'$ are contained within $\mathcal{A}$. Moreover, by the definition of $X^\star = \operatorname{argmax}_{x \in \mathcal{A}} u(x)$, any $x' \in X'$ that achieves the same utility as $x^\star \in X^\star$ must also belong in $X^\star$. Therefore, $X' \subseteq X^\star$.

[C2] **Claim: $X$ forms a closed interval.** We prove our claim by showing that utility function $u$ is concave and continuous. Then we use the property that the $\operatorname{argmax}$ of a concave and continuous function over a closed interval is itself a closed interval.

> (a) **Claim: $u$ is a continuous, concave function.** From Assumption 1, the reward functions $r_A$ and $r_B$ are concave and continuous. By definition, $u$ is the sum of $r_A$ and $r_B$. Since the sum of two concave functions is concave, $u$ is concave. Moreover, since the sum of two continuous functions is continuous, $u$ is continuous, because addition preserves continuity.

(b) **Claim:** $X$ **forms a closed interval in** $[0, q]$**.** From [H1], $u$ is a continuous and concave function. The argmax of a concave and continuous function over a closed interval is itself a closed interval. Since $u$ is defined over the closed interval $[0, q]$, and $X = \text{argmax}_{x \in Q}\, u(\langle x, q - x \rangle)$, it follows that $X$ must also form a closed interval in $[0, q]$.

$\square$

## C  Deriving Properties of Functions in the Marginal Returns Class

In this section we formally derive bounds within which functions belonging to the Marginal Returns Class must lie (Lemma 7) and provide specific properties of these bounds.

We first define a *secant line* as the line through two points $(x_1, f(x_1))$ and $(x_2, f(x_2))$:

$$s(x) = f(x_0) + \frac{f(x_1) - f(x_0)}{x_1 - x_0}(x - x_0).$$

---

**Lemma 7.** *If function $f$ belongs to the Marginal Returns Class, then the following are lower and upper bounds of $f$ at $x$:*

$$L_f(x) = s_i(x), \quad x \in [x_i, x_{i+1}], \quad i = 0, \ldots, n,$$

$$U_f(x) = \begin{cases} s_1(x), & x \in [x_0, x_1], \\ s_{n-1}(x), & x \in [x_n, x_{n+1}], \\ \min\{s_{i-1}(x), s_{i+1}(x)\}, & x \in [x_i, x_{i+1}], \quad i = 1, \ldots, n - 1. \end{cases} \tag{5}$$

---

*Proof.* We first provide properties of secant lines that will be used to show our final result.

- We first show that a secant line between two points of a concave function forms an upper bound outside the interval between those points.

  **Property 4.** Given a concave function $f(x)$ and a secant line $s(x)$ passing through $(x_0, f(x_0))$ and $(x_1, f(x_1))$ with $x_0 < x_1$, we have $s(x) \geq f(x)$ for $x \notin (x_0, x_1)$.

  *Proof.* Consider the case $x \geq x_1$ (the case $x \leq x_0$ is analogous). Since $f$ is concave,

  $$f(x_1) = f\left(\frac{x - x_1}{x - x_0}x_0 + \frac{x_1 - x_0}{x - x_0}x\right) \geq \frac{x - x_1}{x - x_0}f(x_0) + \frac{x_1 - x_0}{x - x_0}f(x).$$

  Multiplying both sides by $(x - x_0)/(x_1 - x_0)$ and rearranging yields

  $$\frac{x - x_0}{x_1 - x_0}f(x_1) - \frac{x - x_1}{x_1 - x_0}f(x_0) \geq f(x).$$

  The left-hand side is an affine function of $x$ and equals $f(x_0)$ at $x = x_0$ and $f(x_1)$ at $x = x_1$. It is therefore an expression for $s(x)$, hence $s(x) \geq f(x)$ for $x \geq x_1$. $\square$

- Next, we show that the slopes of consecutive secant lines of a concave function are non-decreasing.

  **Property 5.** Given a concave function $f(x)$, let $s_0(x)$ be a secant line passing through $(x_0, f(x_0))$ and $(x_1, f(x_1))$ $(x_0 < x_1)$ with slope $m_0$, and $s_1(x)$ be a second secant line passing through $(x_1, f(x_1))$ and $(x_2, f(x_2))$ $(x_1 < x_2)$ with slope $m_1$. Then we have $m_0 \geq m_1$.

*Proof.* Similar to the proof of Property 4, since $f$ is concave,

$$f(x_1) \geq \frac{x_2 - x_1}{x_2 - x_0} f(x_0) + \frac{x_1 - x_0}{x_2 - x_0} f(x_2).$$

From this we obtain

$$\frac{x_2 - x_1}{x_2 - x_0} (f(x_1) - f(x_0)) \geq \frac{x_1 - x_0}{x_2 - x_0} (f(x_2) - f(x_1)),$$

$$\frac{f(x_1) - f(x_0)}{x_1 - x_0} \geq \frac{f(x_2) - f(x_1)}{x_2 - x_1},$$

which we recognize as $m_0 \geq m_1$. □

Lastly, we use these properties to construct the piecewise lower and upper bounds of $f$.

**Lower bound:** For $x \in [x_i, x_{i+1}]$, the corresponding segment of $s_i(x)$ between $(x_i, f(x_i))$ and $(x_{i+1}, f(x_{i+1}))$ is a chord of $f$. Since $f$ is concave, this chord must lie below $f$.

**Upper bound:** Property 4 ensures that for $x \in [x_i, x_{i+1}]$, any secant line $s_j(x)$, $j \neq i$ is an upper bound on $f(x)$. Taking the minimum of two such secant lines also yields an upper bound. Each segment of $U_f(x)$ falls under one of these cases. (Figure 1 shows graphically that the choices of secant lines in $U_f(x)$ yield the best possible upper bounds from these secants.) □

Lastly, we show that the lower and upper bound functions described in Lemma 7 satisfy properties of continuity, monotonicity, and concavity properties.

---

**Property 6.** Let function $f : [0, q] \to \mathbb{R}$ belong to the Marginal Returns class, and define $L_f : [0, q] \to \mathbb{R}$ and $U_f : [0, q] \to \mathbb{R}$ to be the lower and upper bound functions in Lemma 7. Then, the following is true over $[0, q]$:

[C1] $L_f$ and $U_f$ are continuous and pass through the points $(x_i, f(x_i))$, $i = 1, \ldots, n$.

[C2] $L_f$ and $U_f$ are non-decreasing.

[C3] $L_f$ is concave.

[C4] $U_f$ is locally concave over each interval $[x_i, x_{i+1}]$, $i = 0, \ldots, n$.

---

*Proof.* **Continuity of $L_f(x)$:** At each transition point $x_i$, $i = 1, \ldots, n$, $L_f(x)$ transitions from secant line $s_{i-1}(x)$ to $s_i(x)$. Each such pair of consecutive secant lines also intersects at the point $(x_i, f(x_i))$. Therefore $L_f(x)$ is continuous and passes through the points $(x_i, f(x_i))$, $i = 1, \ldots, n$.

**Continuity of $U_f(x)$:** At each transition point $x_i$, $i = 1, \ldots, n$, $U_f(x)$ transitions from secant line $s_i(x)$ to $s_{i-1}(x)$ (see also Figure 1). Thus, similar to $L_f(x)$, $U_f(x)$ passes through the points $(x_i, f(x_i))$, $i = 1, \ldots, n$, and is continuous at these points. Within each interval $[x_i, x_{i+1}]$, $U_f(x)$ is the minimum of two affine functions, and hence a continuous function. Therefore $U_f(x)$ is continuous over $[0, q]$.

**Non-decrease:** Both $L_f(x)$ and $U_f(x)$ are composed of secant lines $s_i(x)$ passing through $(x_i, f(x_i))$, $(x_{i+1}, f(x_{i+1}))$, where $f$ is a non-decreasing function. Therefore all of the $s_i(x)$ are non-decreasing functions. Together with the continuity of $L_f(x)$ and $U_f(x)$, this implies that they are non-decreasing over $[0, q]$.

**Concavity of $L_f(x)$:** We have established that $L_f(x)$ is a continuous piecewise linear function with pieces given by segments of $s_0(x), \ldots, s_n(x)$. Lemma 5 implies that the slopes of $s_0(x), \ldots, s_n(x)$, i.e., the slopes of the pieces of $L_f(x)$, are non-increasing. Hence $L_f(x)$ is concave.

**Local concavity of $U_f(x)$:** In each interval $[x_i, x_{i+1}]$, $i = 0, \ldots, n$, $U_f(x)$ is either an affine function or the minimum of two affine functions. Both of these are concave functions over $[x_i, x_{i+1}]$. □

# D Proofs for Section 4

Many of the proofs in this section are structured around a sequence of intermediate results, denoted [H1], [H2], ..., or [C1], [C2], ..., each of which establishes a key step in the argument. These claims are referenced by label throughout the proofs.

First, we show that $\hat{X}$ is a closed interval.

> **Lemma 3:** Given Assumption 1, for all $t > 1$, the set $\hat{X}_{t-1}$ is a closed interval in $[0, q]$.

*Proof.* $L_u^t(x)$ **is piecewise linear:** As discussed in Section 4.1, we assume that the existing allocations are indexed in increasing order, $x_0 = 0 < x_1 < \cdots < x_t < x_{t+1} = q$. Denote by $s_0^A, \ldots, s_t^A$ the secant lines used to construct lower and upper bounds on $r_A(x)$, where $s_i^A$ passes through $(x_i, r_A(x_i))$ and $(x_{i+1}, r_A(x_{i+1}))$. For $r_B$, we work with its flipped version $r_B(q - x)$, and index its secant lines $s_0^B, \ldots, s_t^B$ such that $s_i^B$ passes through $(x_i, r_B(q - x_i))$ and $(x_{i+1}, r_B(q - x_{i+1}))$. Then it follows from (3) that the lower bound on utility satisfies

$$L_u^t(x) = s_i^A(x) + s_i^B(x), \quad x \in [x_i, x_{i+1}], \quad i = 0, \ldots, t,$$

i.e., it is also piecewise linear with knots $x_0, \ldots, x_{t+1}$ because the knots of the individual lower bounds $L_{r_A}^t(x)$ and $L_{r_B}^t(q - x)$ are aligned.

**Concavity of $L_u^t(x)$ and its implications:** Let $m_i^A$, $m_i^B$ denote the slopes of secants $s_i^A$, $s_i^B$. Since $L_{r_A}^t(x)$ and $L_{r_B}^t(q - x)$ are concave, so too is their sum $L_u^t(x)$, and the slopes $m_i^A + m_i^B$ of $L_u^t(x)$ are thus non-increasing. Furthermore, due to concavity, the maximum $\ell_t = \max_{x \in [0,q]} L_u^t(x)$ is achieved at an interval $[x_{i_l}, x_{i_r}]$ bounded by knots $x_{i_l}$, $x_{i_r}$ (which could be the same if the maximum is unique). It follows from this definition of $[x_{i_l}, x_{i_r}]$ that the slope to the left is strictly positive, $m_{i_l-1}^A + m_{i_l-1}^B > 0$, while the slope to the right is negative, $m_{i_r}^A + m_{i_r}^B < 0$.

**Decomposition of $\hat{X}_t$ into three contiguous intervals:** Since $L_u^t(x) = \ell_t$ for $x \in [x_{i_l}, x_{i_r}]$ and $U_u^t(x) \geq L_u^t(x)$, by the definition of $\hat{X}_t$ (4), we have $[x_{i_l}, x_{i_r}] \subseteq \hat{X}_t$. We show below that $\hat{X}_t$ can also include a second interval $[x_{i_r}, x_{i_r}^+]$ where $x_{i_r}^+ < x_{i_r+1}$, but no points greater than $x_{i_r}^+$. By an analogous proof (which we omit), $\hat{X}_t$ can include a third interval $[x_{i_l}^-, x_{i_l}]$ where $x_{i_l}^- > x_{i_l-1}$, but no points less than $x_{i_l}^-$. Taken together, these facts imply that $\hat{X}_t = [x_{i_l}^-, x_{i_r}^+]$, i.e., a single interval as desired.

**Considering the interval $[x_{i_r}, x_{i_r+1}]$:** First, if $x_{i_r} = x_{t+1} = q$, then we can only have $x_{i_r}^+ = q$ (i.e., $[x_{i_r}, x_{i_r}^+]$ is a single point). Suppose then that $x_{i_r} < q$. Property 6 implies that at the knots $x_{i_r}$, $x_{i_r+1}$, the lower and upper bounds coincide with the utility function, $L_u^t(x_{i_r}) = u(x_{i_r}) = U_u^t(x_{i_r})$ and $L_u^t(x_{i_r+1}) = u(x_{i_r+1}) = U_u^t(x_{i_r+1})$. Furthermore, since $x_{i_r}$ is the right endpoint of the interval that maximizes $L_u^t(x)$, we have $L_u^t(x_{i_r}) = U_u^t(x_{i_r}) = \ell_t$ and $L_u^t(x_{i_r+1}) = U_u^t(x_{i_r+1}) < \ell_t$. We also have from Property 6 that $U_u^t(x)$ is locally concave in $[x_{i_r}, x_{i_r+1}]$ (because $U_{r_A}^t(x)$ and $U_{r_B}^t(q - x)$ are locally concave). Local concavity implies that the super-level set $\{x \in [x_{i_r}, x_{i_r+1}] : U_u^t(x) \geq \ell_t\}$ is a single interval $[x_{i_r}, x_{i_r}^+]$, where $x_{i_r}^+ < x_{i_r+1}$. This super-level set is the restriction of $\hat{X}_t$ to $[x_{i_r}, x_{i_r+1}]$, i.e., $\hat{X}_t \cap [x_{i_r}, x_{i_r+1}] = [x_{i_r}, x_{i_r}^+]$.

**Considering $x \geq x_{i_r+1}$:** From the previous paragraph, $U_u^t(x_{i_r+1}) < \ell_t$. Thus, if $U_u^t(x)$ is non-increasing for $x \geq x_{i_r+1}$, then $\hat{X}_t$ cannot include any of these points. Let us consider first the interval $[x_{i_r+1}, x_{i_r+2}]$. On this interval, the form (5) of the upper bounds $U_{r_A}^t(x)$, $U_{r_B}^t(q - x)$ implies that the slope of $U_u^t(x) = U_{r_A}^t(x) + U_{r_B}^t(q - x)$ can take one of at most four values:

$$\{m_{i_r}^A + m_{i_r}^B, m_{i_r}^A + m_{i_r+2}^B, m_{i_r+2}^A + m_{i_r}^B, m_{i_r+2}^A + m_{i_r+2}^B\}.$$

Since $m_i^A$, $m_i^B$ are slopes of secants of concave functions $r_A(x)$, $r_B(q - x)$, the largest of these four sums is $m_{i_r}^A + m_{i_r}^B$. It was previously shown in this proof that $m_{i_r}^A + m_{i_r}^B < 0$ (by definition of $x_{i_r}$), and thus the slope of $U_u^t(x)$ on $[x_{i_r+1}, x_{i_r+2}]$ must be negative. By similar arguments, the slope of $U_u^t(x)$ on subsequent intervals $[x_{i_r+2}, x_{i_r+3}]$, etc., must be at least as negative. We conclude that $U_u^t(x)$ is indeed non-increasing for $x \geq x_{i_r+1}$. □

In the rest of this section, we additionally use the following notation:

$$d_1^t(x) = L_{h_A}^t(x) - U_{h_B}^t(q - x), \quad d_2^t(x) = L_{h_B}^t(q - x) - U_{h_A}^t(x)$$
$$D_1^t = \{x \in [0, q] : |d_1^t(x)| \leq G\}, \quad D_2^t = \{x \in [0, q] : |d_2^t(x)| \leq G\}.$$

**Property 1:** Given Assumption 1, if an allocation $x$ is contained in the set $D_1^t \cap D_2^t$ at round $t$, then it is fair.

*Proof.* We use proof by contradiction to show that $x \in D_1^t \cap D_2^t$ implies $x$ satisfies EoI.

For contradiction, assume $x \in D_1^t \cap D_2^t$ but does not satisfy EoI. By the definition of EoI, this means $|h_A(x) - h_B(q - x)| > G$. Using the properties of absolute value, this implies that $x$ satisfies at least one of the following conditions:

$$h_A(x) - h_B(q - x) > G, \tag{6}$$
$$h_B(q - x) - h_A(x) > G. \tag{7}$$

First, consider the case where $x$ satisfies (6). By Lemma 7, at round $t$, the values of $h_A(x)$ and $h_B(q - x)$ are guaranteed to lie within the intervals $[L_{h_A}^t(x), U_{h_A}^t(x)]$ and $[L_{h_B}^t(q - x), U_{h_B}^t(q - x)]$, respectively.

Since $x \in D_1^t \cap D_2^t$, we know:

$$|L_{h_A}^t(x) - U_{h_B}^t(q - x)| \leq G \quad \text{and}$$
$$|L_{h_B}^t(q - x) - U_{h_A}^t(x)| \leq G.$$

However, if $x$ satisfies (6), we would have:

$$U_{h_A}^t(x) - L_{h_B}^t(q - x) > G.$$

This creates a contradiction because the possible values of $h_A(x)$ and $h_B(q - x)$, constrained by their respective intervals, cannot result in a difference greater than $G$. Therefore, $x$ cannot satisfy (6).

Similarly, if $x$ satisfies (7), the same argument applies. In both cases, we reach a contradiction, meaning $x$ must satisfy EoI. Therefore, any $x \in D_1^t \cap D_2^t$ is fair.

$\square$

Next, towards showing Lemma 4 and later results, we show that $D_1$ and $D_2$ are empty or closed intervals.

**Property 7.** Given Assumption 1, sets $D_1$ and $D_2$ are empty or closed intervals over $[0, q]$.

*Proof.* At a high level, we show that these sets are empty or closed intervals by showing that $d_1$ and $d_2$ are continuous, monotone functions, and appealing to Lemma 6.

[H1] *Claim: $d_1^t$ and $d_2^t$ are continuous functions over $[0, q]$.* By Property 6, the upper and lower bounds of functions belonging to the Marginal Returns Class are continuous. By Assumption 1, the impact functions belong to this class and employ these bounds. Because $d_1^t$ is the difference of two continuous functions, it is also continuous, because subtraction preserves continuity. The same argument applies to $d_2^t$.

[H2] *Claim: $d_1^t$ and $d_2^t$ are monotone functions over $[0, q]$.* To prove this claim, we show that $d_1^t$ is the sum of two non-decreasing functions, implying it is non-decreasing and therefore monotone. A similar argument holds for $d_2^t$. By definition,

$$d_1^t(x) = L_{h_A}^t(x) - U_{h_B}^t(q - x)$$
$$d_2^t(x) = L_{h_B}^t(q - x) - U_{h_A}^t(x),$$

where all aforementioned bound functions are non-decreasing (by Property 6).

To prove that $d_1^t(x)$ is a non-decreasing function over $[0, q]$, we analyze the behavior of its components. Since $U_{h_B}^t(x)$ is non-decreasing in $x$, $U_{h_B}^t(q - x)$ is non-increasing in $x$. For $x_1 \leq x_2$, we have $q - x_1 \geq q - x_2$. So,

$$U_{h_B}^t(q - x_1) \geq U_{h_B}^t(q - x_2).$$

Multiplying a non-increasing function by $-1$ makes it non-increasing, so $-U_{h_B}^t(q - x)$ is a non-decreasing function of $x$. Because the sum of two non-decreasing functions is non-decreasing, we have that $d_1^t$ is non-decreasing in $x$, and therefore monotone.

A similar argument can be made to show that $d_2^t$ is a non-increasing function. That is, $L_{h_B}^t(q - x)$ is a non-increasing function of $x$, and $-U_{h_A}^t(x)$ is a non-increasing function of $x$. Therefore, their sum is non-increasing, and $d_2^t$ is therefore monotone.

[H3] From [H1] and [H2], functions $d_1^t$ and $d_2^t$ are continuous and monotone. Then from Lemma 6, the sets $\{x \in [0, q] : |d_1^t(x)| \leq G\}$ and $\{x \in [0, q] : |d_2^t(x)| \leq G\}$ are closed intervals (or possibly empty). These sets are exactly $D_1$ and $D_2$, respectively.

$\square$

Next, we show that the guaranteed fair set is a closed interval.

---

**Lemma 4**: At any round $t > 1$, $A_t^-$ is an empty or closed interval in $[0, q]$.

---

*Proof.* This follows directly from Property 7 and the fact that the intersection of two closed intervals is empty or itself a closed interval. By the definition of the guaranteed fair set and Property 1, $A_t^- = D_1 \cap D_2$. From Property 7, we have that $D_1$ and $D_2$ are empty or closed intervals. The intersection of two closed intervals is itself a closed interval (or empty). Therefore, our claim holds. $\square$

Before proving Lemma 5, we introduce the concept of *overlapping bound functions*, which helps build intuition for the proof. We begin by noting that the functions $h_A(x)$ and $h_B(q - x)$ intersect when $G = 0$. Using this observation, we previously proved Lemma 1 by applying the Intermediate Value Theorem to show that there exists an $x \in [0, q]$ such that $h_q(x) = h_A(x) - h_B(q - x) = 0$. This result implies that at some allocation $x$,

$$h_A(x) = h_B(q - x).$$

In other words, the functions $h_A$ and $h_B$ intersect at least once.

Now consider the bounds on $h_A$ and $h_B$ from Lemma 7. At any round $t$, we are guaranteed that:

$$h_A(x) \in [L_{h_A}^t(x), U_{h_A}^t(x)] \quad \text{and} \quad h_B(q - x) \in [L_{h_B}^t(q - x), U_{h_B}^t(q - x)].$$

Since $h_A(x)$ and $h_B(q - x)$ must intersect at some allocation $x$, their respective bound functions $[L_{h_A}^t(x), U_{h_A}^t(x)]$ and $[L_{h_B}^t(q - x), U_{h_B}^t(q - x)]$ must also overlap at some point. To capture the allocations at round $t$ where this overlap occurs, we define the overlap set:

$$D_{\text{overlap}} = \{x \in [0, q] : [L_{h_A}^t(x), U_{h_A}^t(x)] \cap [L_{h_B}^t(q - x), U_{h_B}^t(q - x)] \neq \emptyset\}.$$

Next, we show that the overlap set is a closed interval.

**Property 8.** $D_{overlap}$ is a non-empty, closed interval in $[0, q]$.

*Proof.* Define

$$I_A(x) = [L_{h_A}^t(x), U_{h_A}^t(x)],$$
$$I_B(q - x) = [L_{h_B}^t(q - x), U_{h_B}^t(q - x)].$$

[C1] *Claim: $D_{\text{overlap}}$ is non-empty.* Consider an allocation $x$ where $h_A(x) = h_B(q - x)$. Such an allocation must exist because by Assumption 1, $h_A(x)$ is non-decreasing in $x$ and $h_B(q - x)$ is non-increasing in $x$, and $h_A(0) = h_B(0) = 0$. Therefore, the two impact functions must intersect at some point.

By Lemma 7, the bound functions are guaranteed to contain their respective impact functions at any allocation in $[0, q]$. Therefore, it must be the case that $I_A(x) \cap I_B(q - x) \neq 0$. By definition this $x$ must belong to $D_{\text{overlap}}$.

[C2] *Claim: $D_{\text{overlap}}$ is a closed interval.* We first show that for any $x_1 \leq x_2 \leq x_3$ such that $x_1$ and $x_2$ are in $D_{\text{overlap}}$, $x_2 \in D_{\text{overlap}}$. We then show that $D_{\text{overlap}}$ contains its boundary points.

[H1] Recall from Lemma 7 and Property 6 that the following hold:
- Bound functions $L_{h_A}(x)$ and $U_{h_A}(x)$ are non-decreasing in $x$.
- Bound functions $L_{h_B}(q-x)$ and $U_{h_B}(q-x)$ are non-increasing in $x$.

Notice that $x \in D_{\text{overlap}}$ is equivalently,

$$L_{h_A}(x) \leq U_{h_B}(q-x) \text{ and } L_{h_B}(q-x) \leq U_{h_A}(x).$$

Assume that $x_1 \in D_{\text{overlap}}$ and $x_3 \in D_{\text{overlap}}$ with $x_1 \leq x_3$. Our goal is to show that every $x_2$ such that $x_1 \leq x_2 \leq x_3$ is also contained in $D_{\text{overlap}}$. To confirm this relationship, we need to show that

$$L_{h_A}(x_2) \leq U_{h_B}(q-x_2) \text{ and } L_{h_B}(q-x_2) \leq U_{h_A}(x_2).$$

Consider the first inequality. We derive this expression using given information.

$$L_{h_A}(x_2) \leq L_{h_A}(x_3) \quad \text{[defn of non-decreasing function]}$$
$$U_{h_B}(q-x_3) \leq U_{h_B}(q-x_2) \quad \text{[defn of non-increasing function]}$$
$$L_{h_A}(x_3) \leq U_{h_B}(q-x_3) \quad \text{[true because } x_3 \in D_{\text{overlap}}]$$
$$L_{h_A}(x_2) \leq L_{h_A}(x_3) \leq U_{h_B}(q-x_3) \leq U_{h_B}(q-x_2)$$
$$\implies L_{h_A}(x_2) \leq U_{h_A}(q-x_2).$$

Next, we consider the second inequality. We also derive this expression using given information.

$$U_{h_A}(x_1) \leq U_{h_A}(x_2) \quad \text{[defn of non-decreasing function]}$$
$$L_{h_B}(q-x_2) \leq L_{h_B}(q-x_1) \quad \text{[defn of non-increasing function]}$$
$$L_{h_B}(q-x_1) \leq U_{h_A}(x_1) \quad \text{[true because } x_1 \in D_{\text{overlap}}]$$
$$L_{h_B}(q-x_2) \leq L_{h_B}(q-x_1) \leq U_{h_A}(x_1) \leq U_{h_A}(x_2)$$
$$\implies L_{h_B}(q-x_2) \leq U_{h_A}(x_2).$$

So $x_1$ and $x_3$ in $D_{\text{overlap}}$ imply $x_2 \in D_{\text{overlap}}$.

[H2] Next, we must show that $D_{\text{overlap}}$ contains its boundary points. Note that any non-empty subset of $[0, q]$ has a finite infimum and supremum, which we denote as

$$x_L = \inf(D_{\text{overlap}}), \quad x_U = \sup(D_{\text{overlap}}).$$

Our goal is to show that $x_L$ and $x_U$ both belong to $D_{\text{overlap}}$. For contradiction, assume $x_L \notin D_{\text{overlap}}$. Then at $x_L$, at least one overlap inequality fails (e.g. $L_{h_A}(x_L) > U_{h_B}(q-x_L)$). Because $L_{h_A}(x)$ is non-decreasing in $x$ and $U_{h_B}(q-x)$ is non-increasing in $x$, the same strict inequality $L_{h_A}(x) > U_{h_B}(q-x)$ holds for all $x$ in some interval $(x_L, x_L + \varepsilon)$, leaving no points of $D_{\text{overlap}}$ arbitrarily close to $x_L$ from the right. But this contradicts $x_L$ being the infimum of a nonempty set. Therefore $x_L \in D_{\text{overlap}}$.

We can use a similar argument shows $x_R \in D_{\text{overlap}}$: if $x_R \notin D_{\text{overlap}}$, then either $L_{h_A}(x_R) > U_{h_B}(q-x_R)$ or $LB(\beta) > UA(\beta)$, and by monotonicity that failure also persists in a left-neighborhood of $x_R$, contradicting $x_R$ as the supremum. Therefore $x_L$ and $x_R$ are contained in $D_{\text{overlap}}$.

$\square$

---

**Lemma 5:** At any round $t > 1$, the potential set $A_t^+$ is a closed interval in $[0, q]$.

---

*Proof.* At a high level, we show that the set of potentially fair allocations $A_t^+$ forms a single closed interval by demonstrating that the underlying function defining this set—the minimum possible impact difference $d(x)$—is continuous and quasi-convex. This ensures its sublevel sets are intervals.

The potential fair set at round $t$, denoted $A_t^+$, contains all allocations $x \in [0, q]$ (where $q$ is the total resource budget) for which there could exist an outcome satisfying the Equality of Impact (EoI) condition, given the current information about the impact functions. Specifically, $A_t^+ = \{x \in$

$[0, q] : d(x) \leq G\}$, where $G \geq 0$ is the fairness tolerance parameter from Definition 2 (Equality of Impact). The function $d(x)$ represents the minimum possible absolute difference between a potential impact on group $A$, $y_A$, and a potential impact on group $B$, $y_B$. These potential impacts are constrained by intervals derived from the observed data: $y_A \in I_A(x) = [L_{h_A}^t(x), U_{h_A}^t(x)]$ if group $A$ is allocated $x$, and $y_B \in I_B(x) = [L_{h_B}^t(q - x), U_{h_B}^t(q - x)]$ if group $B$ is allocated $q - x$. Here, $L_{h_i}^t$ and $U_{h_i}^t$ are the lower and upper bounds for the true impact function $h_i$ of group $i \in \{A, B\}$ at round $t$, as constructed from sampled outcomes according to Lemma 7. Thus, $d(x) = \min\{|y_A' - y_B'| : y_A' \in I_A(x), y_B' \in I_B(x)\}$. This minimum difference can be expressed explicitly as:

$$d(x) = \max\left(0, L_{h_A}^t(x) - U_{h_B}^t(q - x), L_{h_B}^t(q - x) - U_{h_A}^t(x)\right).$$

Let $f_1(x) = L_{h_A}^t(x) - U_{h_B}^t(q - x)$ and $f_2(x) = L_{h_B}^t(q - x) - U_{h_A}^t(x)$.

By Property 6 (specifically, properties [C1] Continuity and [C2] Non-decreasing), the functions $L_{h_A}^t(x)$ and $U_{h_A}^t(x)$ are continuous and non-decreasing on $[0, q]$. Similarly, $L_{h_B}^t(y)$ and $U_{h_B}^t(y)$ are continuous and non-decreasing with respect to their argument $y$. Consequently:

- $U_{h_B}^t(q - x)$ is continuous and non-increasing in $x$ on $[0, q]$.

- $L_{h_B}^t(q - x)$ is continuous and non-increasing in $x$ on $[0, q]$.

Therefore:

- $f_1(x) = L_{h_A}^t(x) - U_{h_B}^t(q - x)$ is the sum of a continuous non-decreasing function ($L_{h_A}^t(x)$) and another continuous non-decreasing function ($-U_{h_B}^t(q - x)$). Thus, $f_1(x)$ is continuous and non-decreasing on $[0, q]$.

- $f_2(x) = L_{h_B}^t(q - x) - U_{h_A}^t(x)$ is the sum of a continuous non-increasing function ($L_{h_B}^t(q - x)$) and another continuous non-increasing function ($-U_{h_A}^t(x)$). Thus, $f_2(x)$ is continuous and non-increasing on $[0, q]$.

Since $f_1(x)$ and $f_2(x)$ are continuous, and the $\max$ function preserves continuity, $d(x) = \max(0, f_1(x), f_2(x))$ is continuous on $[0, q]$. As $A_t^+$ is the set $\{x \in [0, q] : d(x) \leq G\}$, it is the sublevel set of a continuous function. Since $d(x) \geq 0$ and $G \geq 0$, $A_t^+$ is the preimage of a closed set (e.g., $[0, G]$ restricted to the range of $d(x)$), and thus $A_t^+$ is a closed set.

To show that $A_t^+$ is a single interval, we examine the shape of $d(x)$.

- As $x$ increases, $f_1(x)$ (non-decreasing) tends to increase from potentially negative to positive values.

- As $x$ increases, $f_2(x)$ (non-increasing) tends to decrease from potentially positive to negative values.

The function $d(x) = \max(0, f_1(x), f_2(x))$ will therefore exhibit the following behavior:

- When $f_2(x)$ is positive and dominates $f_1(x)$ (and 0), $d(x)$ follows $f_2(x)$ and is non-increasing.

- When both $f_1(x) \leq 0$ and $f_2(x) \leq 0$, $d(x) = 0$. This corresponds to the region where the intervals $I_A(x)$ and $I_B(x)$ overlap.

- When $f_1(x)$ is positive and dominates $f_2(x)$ (and 0), $d(x)$ follows $f_1(x)$ and is non-decreasing.

This overall behavior (non-increasing, then potentially constant at zero, then non-decreasing) characterizes $d(x)$ as a quasi-convex function on the interval $[0, q]$.

A key property of such a quasi-convex function $d(x)$ (which we also established is continuous) is that its sublevel sets are convex. Specifically, the definition of quasi-convexity ($d(\lambda x_1 + (1 - $

$\lambda)x_2) \leq \max(d(x_1), d(x_2))$ for $\lambda \in [0,1]$ and $x_1, x_2 \in [0,q]$) ensures that if $d(x_1) \leq c$ and $d(x_2) \leq c$ for some constant $c$, then it must also be that $d(\lambda x_1 + (1-\lambda)x_2) \leq c$. This means that if any two points $x_1, x_2$ belong to a sublevel set $\{x \in [0,q] : d(x) \leq c\}$, then all points on the segment connecting $x_1$ and $x_2$ also belong to that set, which is the definition of a convex set. Since $A_t^+ = \{x \in [0,q] : d(x) \leq G\}$ is such a sublevel set, and $d(x)$ is continuous, $A_t^+$ is both closed and convex. A closed and convex subset of the real line $\mathbb{R}$ (like $A_t^+$ within $[0,q]$) must be a single closed interval (this interval could be empty, a single point, or an interval of the form $[a,b]$).

Thus, $A_t^+$ is a single closed interval in $[0,q]$. $\qquad\square$

Lastly, we show that regardless of how Algorithm 1 chooses allocations, $\hat{X}$ and $A^+$ do not increase in size at each round.

**Property 9.** Given Assumption 1, for all $t > 0$, $A_{t-1}^+ \supseteq A_t^+$.

*Proof.* By definition,

$$A_t^+ = \{x : \exists y_A \in [L_{h_A}^t(x), U_{h_A}^t(x)], y_B \in [L_{h_B}^t(q-x), U_{h_B}^t(q-x)] \text{ such that } |y_A - y_B| \leq G\}.$$

The lower and upper bounds evolve as follows: for all $i \in \{A, B\}$

$$L_{h_i}^t(x) \geq L_{h_i}^{t-1}(x) \quad U_{h_i}^t(x) \leq U_{h_i}^{t-1}(x).$$

Then the intervals $[L_{h_i}^t(x), U_{h_i}^t(x)]$ are nested at $t-1$, i.e., $[L_{h_i}^t(x), U_{h_i}^t(x)] \subseteq [L_{h_i}^{t-1}(x), U_{h_i}^{t-1}(x)]$. Therefore, any pair $(y_A, y_B)$ that exists at round $t$ must also have existed at round $t-1$. Therefore $A_t^+ \subseteq A_{t-1}^+$. $\qquad\square$

**Property 10.** Given Assumption 1, for all $t > 0$, $\hat{X}_t \subseteq \hat{X}_{t-1}$.

*Proof.* For contradiction, assume $\hat{X}_t \not\subseteq \hat{X}_{t-1}$, i.e., there exists an allocation $x$ contained in $\hat{X}_t$ but not in $\hat{X}_{t-1}$. Let $\ell(t) = \max_{x' \in [0,q]} L_u^t(x')$. Then by definition,

$$\hat{X}_t = \{x' : U_u^t(x') \geq \ell(t)\}$$
$$\hat{X}_{t-1} = \{x' : U_u^{t-1}(x') \geq \ell(t-1)\}.$$

The following expressions must hold:

$$U_u^{t-1}(x) \geq U_u^t(x) \quad \text{[monotonicity of the upper bound]}$$
$$U_u^t(x) \geq \ell(t) \quad \text{[defn of } \hat{X}_t]$$
$$\ell(t) \geq \ell(t-1). \quad \text{[monotonicity of the lower bound]}$$

Therefore, $U_u^{t-1}(x) \geq \ell(t-1)$, implying $x \in \hat{X}_{t-1}$. But this contradicts our assumption. So $x$ must also be contained in $\hat{X}_{t-1}$, and therefore $\hat{X}_t \subseteq \hat{X}_{t-1}$. $\qquad\square$

## E  Fair Regret Analysis

In this section we prove Theorem 3, which is restated below.

> **Theorem 3:** Given Assumptions 1 and 2, Algorithm 1 achieves bounded fairness regret. In particular, the cumulative fairness regret satisfies $\mathcal{R}_{\text{fair}}(T) = O(1)$.

*Proof.* To show that $\mathcal{R}_{\text{fair}}(T) = O(1)$, we note that at each round $t$, an allocation decision $x_t$ is located in one of three (mutually exclusive) sub-intervals in $A^+$, and derive the worst-case instantaneous regret in each sub-interval. We then show that the cumulative regret of this worst-case problem is bounded by a constant.

We first show that each allocation made by Algorithm 1 must lie in the potential fair set $A^+$.

**Property 11.** At each round $t$, Algorithm 1 chooses an allocation in $A_t^+$.

*Proof.* Algorithm 1 employs Algorithm 2 to choose an allocation at the next round. Algorithm 2 makes allocation decisions based on three general cases—we show that in each case, the allocation decision resides in $A^+$. In case one (lines two and three of Algorithm 2), $p_L$ is returned. By definition, $p_L \in A^+$. Similarly, in case two (lines four and five), $p_R \in A^+$ is returned. In the last case (lines seven through nine), allocations belonging to $\mathcal{F} = (A^+ \cap \hat{X}) \setminus A^-$ are returned. This definition implies $\mathcal{F} \subseteq A^+$. Therefore, all allocation decisions made by Algorithm 1 belong to $A^+$. □

We can further divide $A^+$ into sub-intervals of interest—specifically, we consider $[p_L, a_L)$, $[a_L, a_R]$, and $(a_R, p_R]$, which form a partition of $A^+$. In each of these cases, we will determine the worst-case instantaneous regret, and use these values later to construct an upper bound on the cumulative regret.

Trivially, note that $x_t \in [a_L, a_R] \implies x_t \in \mathcal{Q}^G$, and no fairness violation occurs because $g(x_t) = 0$ for any $x_t \in \mathcal{Q}^G$. Next, we show that the worst-case regret in $[p_L, a_L)$ and $(a_R, p_R]$ can be obtained at $p_L$ and $p_R$, respectively. In other words, the worst-case regret of $A^+$ can be seen at its boundaries. Towards this, we first provide a property of the function $|h_q(x)| = |h_A(x) - h_B(q - x)|$ that will be used to show this result:

**Lemma 8.** *Given Assumption 1, the function $|h_q(x)| = |h_A(x) - h_B(q - x)|$ is non-increasing over $[0, a_L]$, zero over $[a_L, a_R]$, and non-decreasing over $[a_R, q]$.*

*Proof.* This follows from the properties of the impact functions $h_A$ and $h_B$ defined in the Marginal Return Class (Assumption 1), which implies that $h_A$ is a non-decreasing function of $x$, and $h_B$ is a non-increasing function of $q - x$. We also use the definitions of non-increasing and non-decreasing functions. Consider values $x_1$ and $x_2$ such that $x_1 < x_2$. A function $f$ is non-increasing if $f(x_1) \geq f(x_2)$, and non-decreasing if $f(x_1) \leq f(x_2)$. To prove our claim, we consider each case individually.

- Case: $x \in [0, a_L]$. Note that $h_A$ is non-decreasing in $x$ to $a_L$, and that $h_B$ is non-increasing in $q - x$ to $a_L$, at which point $h_A(a_L) = h_B(q - a_L) = 0$. Then to show that $|h_q|$ is non-increasing, we must show that $\forall x_1$ and $x_2 \in [0, a_L]$, $|h_q(x_1)| \geq |h_q(x_2)|$:

$$|h_q(x_1)| \geq |h_q(x_2)|$$
$$= |h_A(x_1) - h_B(q - x_1)| \geq |h_A(x_2) - h_B(q - x_2)| \quad \text{[defn of } h_q\text{]}$$
$$= h_B(q - x_1) - h_A(x_1) \geq h_B(q - x_2) - h_A(x_2) \quad [h_B > h_A \text{ over } [0, a_L]]$$
$$= h_B(q - x_1) - h_B(q - x_2) \geq h_A(x_1) - h_A(x_2) \quad \text{[rearrange]}.$$

  The last expression holds from the fact that $h_B$ is non-increasing (and therefore the difference $h_B(q - x_1) - h_B(q - x_2)$ is non-negative), and $h_A$ is non-decreasing (and therefore the difference $h_A(x_1) - h_A(x_2)$ is non-positive).

- Case: $x \in [a_L, a_R]$. By Theorem 1, $\mathcal{Q}^G$ is a closed interval and is represented by $[a_L, a_R]$. By definition, for any $x \in \mathcal{Q}^G$, $|h(x)| = 0$.

- Case: $x \in [a_R, q]$. A similar argument as in the first case can be derived, where instead we show that $|h_q(x_1)| \leq |h_q(x_2)|$, and use the fact that $h_A > h_B$ over $[a_R, q]$ to prove this case.

□

Next, we show that over the sub-intervals $[p_L, a_L)$ and $(a_R, p_R]$, the highest fair regret within each sub-interval exists at $p_L$ and $p_R$, respectively. We will use this to

**Property 12.** Given Assumption 1, let $x_1$ and $x_2$ belong to $[0, q]$ such that $x_1 < x_2$. Then

1. If $x_1$ and $x_2$ belong to $[p_L^t, a_L)$, then $g(x_1) \geq g(x_2)$

2. If $x_1$ and $x_2$ belong to $(a_R, p_R^t]$, then $g(x_2) \geq g(x_1)$.

*Proof.* We start by showing the first case. By Lemma 8, $|h|$ is non-increasing over $[0, a_L]$ and therefore non-increasing over $[p_L^t, a_L)$. This implies that $|h_q(x_1)| \geq |h_q(x_2)|$. Then the following inequalities must hold:

$$|h_q(x_1)| \geq |h_q(x_2)| \quad \text{[def of non-increasing function]}$$
$$\implies |h_q(x_1)| - G \geq |h_q(x_2)| - G$$
$$\implies \max(0, |h_q(x_1)| - G) \geq \max(0, |h_q(x_2)| - G)$$
$$= g(x_1) \geq g(x_2). \quad \text{[def of fair regret]}$$

A similar argument can be made to show the second case. By Lemma 8, $|h_q|$ is non-decreasing over $[a_R, q]$ and therefore non-decreasing over $(a_R, p_R^t]$. This implies that $|h_q(x_2)| \geq |h_q(x_1)|$ for all $x_t \in (a_R, p_R^t]$. A similar series of algebraic steps can be used to show $g(x_2) \geq g(x_1)$. $\qquad\square$

By definition, if $\mathcal{R}_{\text{fair}}(T) = O(1)$, then there must exist a positive constant $C$ such that

$$\sum_{t=1}^{T} g(x_t) \leq C,$$

where $R_{\text{fair}}(T) = \sum_{t=1}^{T} g(x_t)$. We split this summation into the partition of $A^+$ that was previously defined. Let $\mathcal{X}_t$ be the set of all allocation decisions made by Algorithm 1 from round 1 until (and including) round $t$, and let

$$P_L(t) = \mathcal{X}_t \cap [0, a_L) \quad P_R(t) = \mathcal{X}_t \cap (a_R, q].$$

The cumulative regret can then be split into the sum of the regret due to allocations belonging to these sets:

$$\sum_{t=1}^{T} g(x_t) = \left( \sum_{x \in P_L(T)} g(x) \right) + \left( \sum_{x \in P_R(T)} g(x) \right) + \underbrace{\left( \sum_{x \in \mathcal{Q}^G} g(x) \right)}_{\text{allocations in fair set}}$$

$$= \left( \sum_{x \in P_L(T)} g(x) \right) + \left( \sum_{x \in P_R(T)} g(x) \right) + 0.$$

Property 12 implies that choosing the boundary of $A^+$ at each round leads to an upper bound on the cumulative regret by summing an upper bound on the instantaneous regret:

$$\sum_{t=1}^{T} g(x_t) = \left( \sum_{x \in P_L(T)} g(x) \right) + \left( \sum_{x \in P_R(T)} g(x) \right)$$

$$\leq \left( \sum_{x_t \in P_L(T)} g(p_L^t) \right) + \left( \sum_{x_t \in P_R(T)} g(p_R^t) \right).$$

A valid but looser upper bound can be derived by summing the maximum possible regret from both $P_L$ and $P_R$ for every round $t = 1, \cdots, T$:

$$\sum_{t=1}^{T} g(x_t) \leq \sum_{t=1}^{T} g(p_L^t) + \sum_{t=1}^{T} g(p_R^t). \tag{8}$$

In the rest of this proof, we derive a constant to bound each term in (8). We first focus (in bullets [H1] to [H4]) on the left boundary of $A^+$, and derive a constant that upper-bounds $\sum_{t=1}^{T} g(p_L^t)$. In [H1], we derive an expression for $p_L$ in terms of the impact function bounds. In [H2] and [H3], we derive an upper bound on the worst-case instantaneous regret per round. Lastly, in [H4], we use this derivation to derive a constant that upper-bounds $\sum_{t=1}^{T} g(p_L^t)$.

[H1] **Derive an expression for $p_L$ using the impact function bounds.** Denote $x_i$ as the leftmost neighbor of $p_L^t$. Let $x_{\text{left}}$ represent the sample left of $x_i$. If no such allocation exists, let $x_{\text{left}} = 0$ and $h_A(x_{\text{left}}) = 0$. Similarly, let $x_{\text{right}}$ represent the sample right of $x_i$, and denote $x_{\text{right}} = 0$ if none exists. The left boundary of $A_t^+$ can then be defined as the $x$-value satisfying

$$L_{h_B}^t(q - x) - U_{h_A}^t(x) = G, \tag{9}$$

where $x \in [x_i, p_L]$ and $L_{h_B}^{t_0}(q - x)$ and $U_{h_A}^t(x)$ are defined as

$$L_B^t(q - x) = \frac{h_B(q - x_i) - h_B(q - x_{\text{right}})}{(q - x_i) - (q - x_{\text{right}})}((q - x) - (q - x_{\text{right}})) + h_B(q - x_i)$$

$$= \underbrace{\frac{h_B(q - x_i) - h_B(q - x_{\text{right}})}{x_{\text{right}} - x_i}}_{m_L}(x_{\text{right}} - x) + h_B(q - x_i)$$

$$U_A^t(x) = \underbrace{\frac{h_A(x_{\text{left}}) - h_A(x_i)}{(x_{\text{left}} - x)}}_{m_U}(x - x_i) + h_A(x_i).$$

As the round number increases, $x_i$ and $x_{\text{left}}$ can be replaced with $x_t$ and $x_{t-1}$, respectively. We adopt this notation throughout the remainder of this proof.

[H2] **Derive a lower bound on the minimal change in $p_L$.** In [H1], we derived an expression for $p_L$ based on the bounds of the impact functions. In this section, we use Property 9, which states that $A_{t+1}^+ \subseteq A_t^+$, to derive a lower-bound on the smallest amount that $p_L$ can change between consecutive rounds. In [H3] we show that this implies a worst-case fair instantaneous regret incurred between rounds.

Recall that $x = p_L$ is the allocation that satisfies $L_{h_B}(q - x) - U_{h_A}(x) = G$. Intuitively, when the slopes of $L_h^t$ and $U_h^t$ become steeper, $p_L^{t+1}$ shifts to the left, closer to $p_L^t$. We formally establish this (general) relationship in the following property.

**Property 13.** Define the following lines, where $\epsilon > 0$, $m_l$ and $m_u$ are non-negative values, and $b_u < b_l$:

$$L(x) = -m_l x + b_l, \quad U(x) = m_u x + b_u,$$
$$L'(x) = -(m_l + \epsilon)x + b_l, \quad U'(x) = (m_u + \epsilon)x + b_u.$$

Increasing the steepness of either $L(x)$ or $U(x)$ shifts the intersection point further to the left. Specifically, let $x_{LU}$ denote the intersection of $L$ and $U$. Then the intersection of $L$ and $U'$ occurs at a value $x < x_{LU}$, and the intersection of $L'$ and $U$ also occurs at a value $x < x_{LU}$.

*Proof.* The intersection of $L$ and $U$ is the solution to $-m_l x + b_l = m_u x + b_u$, which gives $x_{LU} = \frac{b_l - b_u}{m_l + m_u}$.

The intersection of $L$ and $U'$ is the solution to $-m_l x + b_l = (m_u + \epsilon)x + b_u$, which gives $x_{LU'} = \frac{b_l - b_u}{m_l + m_u + \epsilon}$.

Since $\epsilon > 0$, the denominator of $x_{LU'}$ is greater than the denominator of $x_{LU}$. Because the numerator $(b_l - b_u)$ is positive, it follows that $x_{LU'} < x_{LU}$.

A similar argument holds for the intersection of $L'$ and $U$, which gives $x_{L'U} = \frac{b_l - b_u}{m_l + m_u + \epsilon}$, also less than $x_{LU}$. Therefore, increasing the steepness of either line shifts the intersection point to the left. $\square$

In context of the impact functions, Property 13 implies that by increasing the steepness of slopes $m_U$ and $m_L$, the intersection of the bounds $L_{h_B}(q - x)$ and $U_{h_A}(x)$ moves leftward. Therefore, making $m_L$ and $m_U$ as steep as possible results in a lower bound on the smallest difference between the allocations chosen in consecutive rounds, and therefore the worst possible regret at the next round.

We can find the lines representing the an upper bound on $m_U$ and $m_L$ using the definition of concavity. Define the lines passing through the origin and the impact functions at $x_i$ as

$$L_B^i(q - x) = \frac{h_B(q - x_i)}{(q - x_i)}(q - x) \tag{10}$$

$$U_A^i(x) = \frac{h_A(x_i)}{x_i}x. \tag{11}$$

Then concavity implies that the impact of any allocation $x \geq x_i$ is bounded above by $U_A^i(x)$, and that the impact of any allocation $q - x \leq q - x_i$ is bounded below by $L_B^i(q - x)$. We can therefore substitute (11) and (10) into (9) and solve for the allocation resulting in the smallest difference between consecutive rounds. We solve for $q - x_t$ as this has been more intuitive for showing our main result. The final form is

$$q - x_t = (q - x_{t-1}) \left[ 1 - \frac{x_{t-1}g(x_{t-1})}{x_{t-1}h_B(q - x_{t-1}) + (q - x_{t-1})h_A(x_{t-1})} \right]. \tag{12}$$

Lastly, in Lemma 9, we show that $q - x_t < q - x_{t-1}$, i.e., the samples (for group $B$) are strictly decreasing towards $q - a_L$ when the allocation is unfair.

**Lemma 9.** *Given Assumption 1, for all $t$, $q - x_t < q - x_{t-1}$.*

*Proof.* Consider the iterative relationship defined in (12), which determines each subsequent allocation. To analyze the behavior of this sequence, we define the function $f$ so that $x_t = f(x_{t-1})$:

$$f(q - x) = (q - x) \left[ 1 - \frac{xg(x)}{xh_B(q - x) + (q - x)h_A(x)} \right].$$

Notice that $f(q-x)$ is equivalent to some fraction of its input, i.e., $f(q-x) = (q-x)[1-c]$. Therefore, if we show that $0 < c < 1$, then this implies $f(q - x_{t-1}) < q - x_{t-1}$.

We first show that $c > 0$. As our goal is to find the worst-case instantaneous fair regret, we assume the input $x$ is unfair. Therefore the function $g$ is always positive. Lastly, by Assumption 1, $h_B$ and $h_A$ are non-negative functions. Therefore, the numerator and denominator are both positive values, and are never zero.

Next, we show that $c < 1$. If this is true, then the denominator must be greater than the numerator, i.e., $xh_B(q - x) + (q - x)h_A(x) > xg(x)$. For contradiction, assume instead that $xh_B(q - x) + (q - x)h_A(x) < xg(x)$. Then the series of simplifications below must hold:

$$xh_B(q - x) + (q - x)h_A(x) < xg(x)$$
$$xh_B(q - x) + (q - x)h_A(x) < {\color{orange}xh_B(q - x) - xh_A(x) - xG} \tag{13}$$
$$xh_B(q - x) + {\color{orange}qh_A(x) - xh_A(x)} < xh_B(q - x) - xh_A(x) - xG$$
$$qh_A(x) < -xG,$$

where $x[h_B(q - x) - h_A(x) - G]$ is substituted for $xg(x)$ in (13). The last expression is a contradiction because $q$ and $G$ are non-negative by definition, and by Assumption 1, $h_A$ is a non-negative function. Therefore, $0 < c < 1$, and for all $t$, $q - x_t < q - x_{t-1}$. $\quad\square$

[H3] **Simplifying the bound.** In this section we show that the bound on the smallest improvement between $p_L^t$ and $p_L^{t+1}$ derived in [H2] implies an upper bound on the instantaneous fair regret between rounds. To show our result, we first show that Assumption 2 implies that the instantaneous fairness regret function $g$ is Lipschitz continuous over $[0, q]$.

**Lemma 10.** *Given Assumptions 1 and 2, the instantaneous regret function $g(x) = \max(0, |h(x)| - G)$ is Lipschitz continuous.*

*Proof.* From Assumption 1, the impact functions $h_A$ and $h_B$ are concave, continuous, and finite-valued over $[0, q]$. From Assumption 2, we have that the one-sided derivatives at 0 and $q$ are well-defined and finite. Let

$$L_{h_A} = \max\{|\partial^+ h_A(0)|, |\partial^- h_A(q)|\}, \quad L_{h_B} = \max\{|\partial^+ h_B(0)|, |\partial^- h_B(q)|\}.$$

Then the impact functions satisfy

$$|h_A(x) - h_A(y)| \leq (L_{h_A} + L_{h_A})|x - y|,$$
$$|h_B(x) - h_B(y)| \leq (L_{h_B} + L_{h_B})|x - y|$$

for all $x, y \in [0, q]$, i.e., the impact functions are Lipschitz continuous. The function $g$ is a 1-Lipschitz transform of $h$ (it does not increase the distance between any two function values), and is therefore Lipschitz continuous with constant $L_h = L_{h_A} + L_{h_B}$. $\qquad\square$

From Lemma 9, we have that the sequence $q - x_t$ is strictly decreasing. The relationship of $q - x_t$ and $q - x_{t-1}$ can be simplified further by employing the Lipschitz continuity of $h_A$ and $h_B$:

$$h_A(x) \leq L_{h_A} x, \qquad h_B(q - x) \leq L_{h_B}(q - x).$$

Then Equation (12) can be simplified as follows:

$$
\begin{aligned}
f(q - x) = (q - x) &\left[ 1 - \frac{xg(x)}{xh_B(q - x) + (q - x)h_A(x)} \right] \\
\leq (q - x) &\left[ 1 - \frac{xg(x)}{xL(q - x) + (q - x)Lx} \right] \\
= (q - x) &\left[ 1 - \frac{g(x)}{2L(q - x)} \right] \\
= (q - x) &- \frac{(q - x)g(x)}{2L(q - x)} \\
= (q - x) &- \frac{1}{2L}g(x),
\end{aligned}
$$

where $L = \max(L_A, L_B)$. Solving for $x_t$, we obtain

$$q - x_t \leq (q - x_{t-1}) - \frac{1}{2L}g(x_{t-1})$$

$$x_t \geq x_{t-1} + \frac{1}{2L}g(x_{t-1}). \tag{14}$$

When $x_{t-1}$ is unfair, this is a strictly increasing, positive sequence. If $x_{t-1}$ is fair, then $a_L$ has been found and the regret incurred is zero.

[H4] **Finding the cumulative fair regret.** Next, we use the relationship in (14) to calculate the cumulative regret when choosing $p_L$ at each round. Rearranging (14), we have that

$$g(x_{t-1}) \leq 2L(x_t - x_{t-1}).$$

This implies that the cumulative regret can be bounded as

$$
\begin{aligned}
\sum_{t=1}^{T} g(p_L^t) &\leq \sum_{t=1}^{T} 2L(x_t - x_{t-1}) \\
&= 2L(x_T - x_0) \quad \text{[simplifying telescoping sum]} \\
&\leq 2L(a_L - 0) \\
&= 2La_L
\end{aligned}
$$

A similar constant can be derived when considering the cumulative regret when choosing $p_R$ at each round. In this case, the relationship between $q - x_t$ and $q - x_{t-1}$ can be written as

$$q - x_{t+1} \geq (q - x_t)\left[ 1 + \frac{x_t g(x_t)}{(q - x_t)h_A(x_t) + x_t h_B(q - x_t)} \right]$$

Using a similar Lipschitz substitution, we can derive a similar expression for the instantaneous fair regret. Rearranging the above expression, we have

$$\left(\frac{q - x_{t+1}}{q - x_t} - 1\right)\left[(q - x_t)h_A(x_t) + x_t h_B(q - x_t)\right] \geq x_t g(x_t)$$

$$\implies x_t g(x_t) \leq \frac{x_t - x_{t+1}}{q - x_t}\left[(q - x_t)h_A(x_t) + x_t h_B(q - x_t)\right] \quad \text{[simplify]}$$

$$\leq \frac{x_t - x_{t+1}}{q - x_t}\left[(q - x_t)Lx_t + x_t L(q - x_t)\right] \quad \text{[Lipschitz substitution]}$$

$$= (x_t - x_{t+1})2x_t L \quad \text{[simplify]}$$

$$\implies g(x_t) \leq (x_t - x_{t+1})2L.$$

Then the cumulative fair regret can be written as

$$\sum_{x_t \in P_R} g(p_R^t) \leq 2L \sum_{t=1}^{T}(x_t - x_{t+1})$$

$$= 2L(x_0 - x_T) \quad \text{[telescoping sum]}$$

$$\leq 2L(q - a_R).$$

Therefore, the total cumulative fair regret $\mathcal{R}_{\text{fair}}(T)$ can be bounded as follows:

$$\sum_{t=1}^{T} g(x_t) \leq \left(\sum_{x_t \in P_L(T)} g(p_L^t)\right) + \left(\sum_{x_t \in P_R(T)} g(p_R^t)\right)$$

$$\leq \sum_{t=1}^{T} g(p_L^t) + \sum_{t=1}^{T} g(p_R^t)$$

$$\leq 2La_L + 2L(q - a_R) = 2L(a_L + (q - a_R)),$$

and therefore Algorithm 1 achieves a cumulative regret of $O(1)$, with the bounding constant $C = 2L(a_L + (q - a_R))$.

$\square$

## F  Empirical Analysis Details

### F.1  Environment Details

The functions used to model the deterministic environment are as follows. In all of the functions below, $e_{h_A}$, $e_{h_B}$, $e_{r_A}$, and $e_{r_B}$ represent noise. In the noiseless setting, these are set to zero when $x = 0$; otherwise, they are drawn from either a uniform or Gaussian distribution.

**Imbalanced-reward environment (IRE).**  In this environment, the reward and impact functions are identical:

$$r_A(x) = 15\log(5x + 1) + e_{r_A}, \quad e_{r_A} \sim \mathcal{N}(0, 0.1)$$

$$r_B(x) = \begin{cases} -w(x - 50)^2 + 2500w + e_{r_B} & \text{if } 0 \leq x < 50 \\ 2500w + e_{r_B} & \text{if } x \geq 50, \end{cases}$$

$$h_A(x) = 15\log(5x + 1) + e_{h_A},$$

$$h_B(x) = \begin{cases} -w(x - 50)^2 + 2500w + e_{r_B} & \text{if } 0 \leq x < 50 \\ 2500w + e_{h_B} & \text{if } x \geq 50, \end{cases}$$

where $w = 0.015$, $e_{h_B} \sim \mathcal{N}(0, 0.1)$, $e_{r_A} \sim \mathcal{N}(0, 0.1)$, $e_{r_B} \sim \mathcal{N}(0, 0.1)$. The allocation satisfying $G = 0$ (the minimum of the red function in the top plot of Figure 4) minimizes disparity. As $G$ increases, notice that the optimal allocation shifts closer to $x^*$.

**Imbalanced-impact environment (IIE).**  In this environment, the reward and impact functions measure separate outcomes, and the impact functions are imbalanced. The reward functions are represented by power functions and the impact functions are represented by log functions.

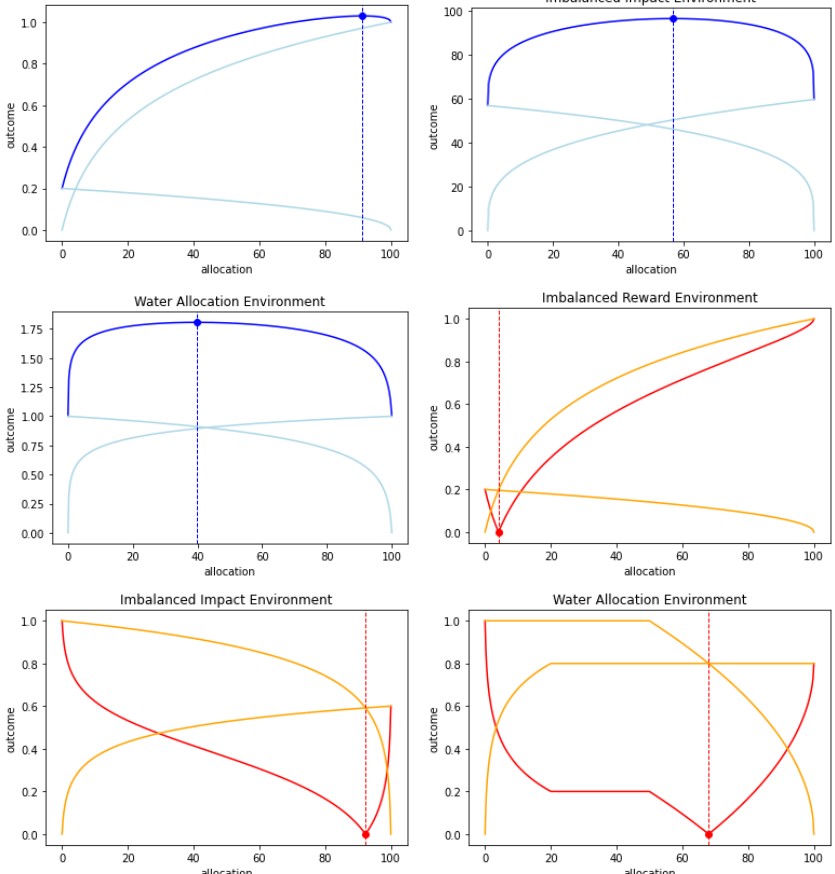

Figure 4: **Upper plots.** The three utility functions (blue) and respective reward functions (light blue) considered in the experimental section. The reward-maximizing allocation is also labeled: 58.52 for IRE, 56.69 for IIE, and for 40.12 SE. **Lower plots.** The three pairs of impact functions considered (yellow), and their respective absolute difference (red). The allocation satisfying EoI when $G = 0$ is labeled: 2.24 for IRE, 90.10 for IIE, and 78.87 for SE. Further function details are in Appendix F.1.

$$r_A(x) = 15x^{0.3} + e_{r_A} \quad e_{h_A} \sim \mathcal{N}(0, 3)$$
$$r_B(x) = 18x^{0.25} + e_{r_B} \quad e_{h_A} \sim \mathcal{N}(0, 3)$$
$$h_A(x) = 7\log(3x + 1) + e_{h_A} \quad e_{h_A} \sim \mathcal{N}(0, 0.1)$$
$$h_B(x) = 10\log(5x + 1) + e_{h_B} \quad e_{h_B} \sim \mathcal{N}(0, 0.4)$$

**Water allocation environment (WAE).** In this environment, the reward and impact functions measure separate outcomes. The impact functions model logarithmic growth, while the reward functions model scenarios where growth is slow, and beyond a certain point, additional allocations yield no further gains.

$$h_A(x) = 3\log(60x + 1) + e_{h_A} \quad e_{h_A} \sim \mathcal{N}(0, 0.2)$$
$$h_B(x) = 4.5\log(3x + 1) + e_{h_B} \quad e_{h_A} \sim \mathcal{N}(0, 0.5)$$
$$r_A(x) = \begin{cases} -w(x - 50)^2 + 2500w + e_{r_A} & \text{if } x < 50 \\ 2500w + e_{r_A} & \text{if } x \geq 50 \end{cases}, \text{where } w = 0.01, e_{r_A} \sim \mathcal{N}(0, 2).$$
$$r_B(x) = \begin{cases} -w(x - 50)^2 + 2500w + e_{r_B} & \text{if } x < 50 \\ 2500w + e_{r_B} & \text{if } x \geq 50 \end{cases}, \text{where } w = 0.015, e_{r_B} \sim \mathcal{N}(0, 1)$$

### F.2 Additional Baseline Details for Deterministic Setting

This section further details the baselines used in the non-noisy setting.

**Explore-then-Commit (ETC)**  This baseline implements the two-phase algorithm described in Section 7. It takes the total time $T$ and an exploration budget $n_{\text{explore}}$ as hyperparameters.

1. **Phase 1: Explore ($t = 1$ to $n_{\text{explore}}$)**
   - The algorithm first defines $n_{\text{explore}}$ bins (strata) by creating $n_{\text{explore}} + 1$ evenly-spaced boundaries.
   - For each bin $i \in [0, \ldots, n_{\text{explore}} - 1]$, it samples exactly one allocation $a_i$ using a uniform random sampler.
   - For each sampled allocation $a_i$, it samples the reward and impact functions, then calculates the total welfare and EoI violation.
   - The algorithm stores all $n_{\text{explore}}$ allocations, welfare values, and violations. These $n_{\text{explore}}$ allocations are the first $n_{\text{explore}}$ actions in the final returned array.

2. **Phase 2: Commit ($t = n_{\text{explore}} + 1$ to $T$)**
   - After the exploration phase, the algorithm analyzes its stored results. It first finds the minimum fairness violation achieved: $v_{\min} = \min(v_0, \ldots, v_{n_{\text{explore}}-1})$.
   - It identifies the set of all indices that achieved this minimum violation: $\mathcal{I}_{\text{fair}} = \{i \mid v_i = v_{\min}\}$.
   - From this "most fair" set, it finds an index corresponding to the single best allocation by maximizing welfare $\hat{a}^\star$.
   - For all remaining $T - n_{\text{explore}}$ rounds, the algorithm "commits" by repeatedly playing $\hat{a}^\star$.

The final output is an array of length $T$, consisting of the $n_{\text{explore}}$ unique sampled allocations followed by $T - n_{\text{explore}}$ allocations $\hat{a}^\star$.

**Brent-Search (BS)**  Simulates an oracle with full knowledge of the problem structure (except how to construct bounds). It operates in two main phases, logging every function evaluation to simulate a sampling budget $T$.

1. **Phase 1: Find Fair Set $\mathcal{Q}^G$**
   - The algorithm uses the problem's structural properties (e.g., Lemma 8) and a numerical root-finder to solve for the exact boundaries of the fair set, $\mathcal{Q}^G = [a_L, a_R]$. This is done by first finding an allocation $a_{G=0}$ such that $|h_A(x) - h_B(q - x)| = 0$. Then, a numerical root-finder is used to find the boundaries of $\mathcal{Q}^G$ over $[0, a_{G=0}]$ and $[a_{G=0}, q]$. All function evaluations (samples) used during the root-finding process are recorded and count toward the allocation budget.

2. **Phase 2: Optimize Reward within $\mathcal{Q}^G$**
   - The algorithm defines the reward function to be maximized: $r(x) = r_A(x) + r_B(q - x)$.
   - If the fair set is a single point (i.e., `np.isclose($a_L$, $a_R$)`), that point is designated the optimal allocation $a^\star$.
   - Otherwise, it calls `scipy.optimize.minimize_scalar` to find the allocation $X$ that maximizes $r(x)$ (by minimizing $-r(x)$) within the bounds $[a_L, a_R]$, using the `'bounded'` (Brent) method.
   - All function evaluations performed by this optimizer are appended to the `allocations` list from Phase 1.

3. **Budget Handling and Commit Phase**
   - At this point, `allocations` is a list of all samples from both stages.
   - If the total number of samples $t_{\text{search}} = \text{len(allocations)}$ is less than $T$, the algorithm "commits" to $a^\star$ for the remaining $T - t_{\text{search}}$ rounds.
   - If $t_{\text{search}} \geq T$, the search process exhausted the budget. The algorithm simply truncates the list and returns the first $T$ allocations it sampled (as a NumPy array).

### F.3 Baselines for Noisy setting

**Evolutionary multi-objective algorithms** For evaluating our noisy algorithm (Algorithm 3), we compare against two common evolutionary multi-objective optimization algorithms: Non-dominated Sorting Genetic Algorithm III (NSGA-III) [Deb and Jain, 2013] and Multi-Objective Evolutionary Algorithm Based on Decomposition (MOEA/D) Zhang and Li [2007]. These are suitable for problems with continuous variables and multiple objectives, such as maximizing total reward while minimizing fairness violations (or satisfying a fairness constraint).

Both NSGA-III and MOEA/D were configured to solve a two-objective problem:

1. **Maximize utility**: $r_A(a_A) + r_B(q - a_A)$
2. **Minimize fairness gap**: $|h_A(a_A) - h_B(q - a_A)|$

The decision variable is $a_A \in [0, q]$, with $a_B = q - a_A$.

- **NSGA-III**: Uses a set of pre-defined reference directions to maintain diversity among solutions on the Pareto front, particularly effective for many-objective problems, but also commonly used for two or three objectives. In our implementation, we used 'get_reference_directions('das-dennis', 2, n_partitions=k)' to generate reference directions, where $k$ was chosen such that the number of reference directions approximately matched the target 'pop_size'. The 'pop_size' and 'n_generations' were tuned to ensure the total number of function evaluations (queries to reward and impact functions) matched the 150-query budget of our algorithm. The problem was defined to output '[-reward, fairness_gap]' to align with minimization objectives. From the resulting Pareto front, solutions where the fairness gap was $\leq G$ were considered feasible. The feasible solution with the highest reward was selected as the 'best' solution for that run. The history of all evaluated individuals across all generations was used to calculate cumulative regret.

- **MOEA/D**: Decomposes the multi-objective optimization problem into a number of single-objective scalar subproblems, which are then solved simultaneously using information from neighboring subproblems. We used the Tchebycheff decomposition method. Reference directions were generated using 'get_reference_directions('das-dennis', 2, n_partitions=k)', where $k$ was chosen to align the number of subproblems (and thus population size) with the target 'pop_size'. The population size was tuned to match the 150-query budget of our algorithm. Crossover was set as Simulated Binary Crossover (SBX) with 'prob=0.9', 'eta=20', and mutation as Polynomial Mutation (PM) with 'prob=0.9', 'eta=10'. Our MOEAD implementation uses default crossover and mutation if not explicitly passed. We also set 'n_neighbors=15' and 'prob_neighbor_mating=0.9'. Similar to NSGA-III, the problem was defined to output '[-reward, fairness_gap]'. Feasible solutions (gap $\leq G$) were identified from the Pareto front, and the one maximizing reward was chosen. The history of all evaluated individuals was used for regret calculation.

For both baselines, the reward ($r_A, r_B$) and impact ($h_A, h_B$) functions provided to them were the noisy versions, meaning each function call returned the true value plus a sample from the noise distribution $\mathcal{N}(0, 0.0577^2)$. The total number of queries (function evaluations) for these baselines was 'n_generations * pop_size', which was set to be comparable to our algorithm's 150 queries.

### F.4 Computational environment

All experiments were conducted on a MacBook Air (M1, 2020 model) featuring an Apple M1 chip (8-core CPU) and 8 GB of RAM, running macOS Sequoia 15.1.1. No external GPUs were utilized. The experiments were implemented in Python 3.10.9. Key libraries utilized include NumPy (version 1.22.4), SciPy (version 1.7.3), Matplotlib (version 3.5.1), and scikit-learn (version 1.4.1.post1), Pymoo (version 0.6.1.3).

### F.5 Computational Efficiency of Algorithm 1

In this section we discuss the computational efficiency of updating the bounds of the reward and impact functions, as well as the set estimators $A^+$, $A^-$, and $\hat{X}$. The key observation is that these

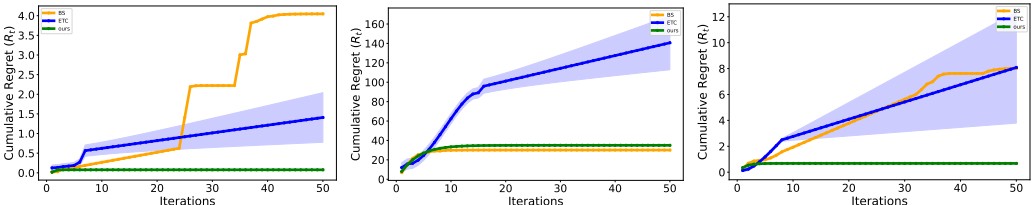

Figure 5: Cumulative reward regret across three environments: WAE, IIE, and IRE (left to right). Each plot shows 50 rounds. For the stochastic `Explore-then-Commit (ETC)` baseline, the shaded region indicates standard deviation over 50 trials. The `Brent-Search (BS)` baseline and Algorithm 1 are deterministic, so only a single trial is shown.

bounds are constructed directly from observed allocations using simple algebraic operations on secant lines. As a result, the updates do not require solving nested optimization problems or performing costly numerical approximations, and the overall procedure remains lightweight. This makes the computational complexity of each round essentially linear in the number of samples, with only modest overhead from maintaining the bound functions.

In addition to updating the bound functions, maintaining the set estimates $A^+$, $A^-$, and $\hat{X}$ is also computationally straightforward. Since each set is a closed interval, it can be represented by just two endpoints, and the update step reduces to simple algebra. Importantly, this process does not require enumerating all possible allocations: in the two-group setting, the maximizers arise at intersection points of neighboring allocations (or at the allocations themselves), which can be identified analytically.

### F.6 Additional Experiments

In this section, we present experiments evaluating the *reward regret* of Algorithm 1. We define cumulative reward regret as the sum of the absolute difference between the optimal allocation $a^\star$ and the allocation chosen at each round:

$$\mathcal{R}_{\text{reward}}(T) = \sum_{t=1}^{T} \left| a^\star - u(a^t) \right|.$$

Figure 5 shows the cumulative reward regret for the baselines and Algorithm 1. While `Brent-Search` exhibits high reward regret in the WAE and IRE environments, it achieves lower regret than Algorithm 1 in the IIE environment. The more information-limited `Explore-then-Commit` baseline consistently performs poorly across all environments.

Overall, Algorithm 1 consistently achieves sublinear cumulative reward regret across all three environments, with the performance gap being particularly pronounced in the WAE and IRE environments.

### F.7 Additional Noisy Algorithm Details

This section provides further details on the Gaussian Process (GP) estimators used in our noisy algorithm (Algorithm 3) in Section 7. Our noisy algorithm utilizes GPs to model the unknown reward $(r_A, r_B)$ and impact $(h_A, h_B)$ functions when observations are subject to noise. Each of these four functions is modeled by an independent GP.

**Kernel selection and hyperparameter tuning**  We employed a systematic approach to select appropriate kernels for each function $(r_A, r_B, h_A, h_B)$ within each of the three environments (IRE, IIE, WAE).

- **Candidate kernels:** We considered three common kernel types suitable for functions exhibiting diminishing marginal returns:
  - Radial Basis Function (RBF) kernel: 'C(1.0, (1e-3, 1e3)) * RBF()'.
  - Matérn kernel: 'C(1.0, (1e-3, 1e3)) * Matern()'.
  - Rational Quadratic kernel: 'C(1.0, (1e-3, 1e3)) * RationalQuadratic()'.

The 'ConstantKernel C' is used to scale the kernel.

- **Tuning process:** For each true underlying function in each environment, we performed a 'GridSearchCV' with 5-fold cross-validation to identify the best kernel and its optimal hyperparameters ('length_scale' for RBF, 'length_scale' and 'nu' for Matérn, 'length_scale' and 'alpha' for Rational Quadratic). This tuning was performed on noise-free samples of the true functions, with 400 training points linearly spaced between 0 and $q = 100$.

- **Storing kernels:** The best-performing kernel (with its optimized hyperparameters) for each function and environment was used during the experiments.

- **Special case:** For the Imbalanced-Rewards Environment (IRE), the reward functions were set to be identical to their respective impact functions ($r_A = h_A, r_B = h_B$), so the kernels for $r_A$ were set to those tuned for $h_A$, and similarly for $r_B$ and $h_B$. An override was also implemented for the 'IR' environment's 'hB' function to use a specific Matern kernel with 'length_scale=0.5', 'nu=2.5', and a lower GP 'alpha=0.001' for better fitting in a specific region.

**GP model configuration in noisy algorithm** When the 'NoisyAlgorithm' is initialized, it creates instances of 'NoisyFairnessFunction' and 'NoisyUtilityFunction'. These classes, in turn, initialize separate 'GaussianProcessRegressor' models from scikit-learn for $h_A, h_B, r_A$, and $r_B$, using the pre-tuned kernels.

- The 'GaussianProcessRegressor' for each function was configured with 'n_restarts_optimizer=10' and an 'alpha=1e-3' (representing the noise variance assumed by the GP model, which can also be interpreted as a Tikhonov regularization term). This 'alpha' in the GP regressor is distinct from the Rational Quadratic kernel's alpha parameter.

**Handling Noisy Observations**

- **Noise model:** In the noisy setting, when the environment is sampled at an allocation $a_t$, the true function values $r_i(a_t)$ and $h_i(a_t)$ are corrupted by additive noise. The noise $\epsilon_{f_i}$ for each function $f \in \{r, h\}$ and group $i \in \{A, B\}$ is drawn independently from a zero-mean normal distribution $\mathcal{N}(0, \sigma^2)$.

- **Noise level:** For the experiments, the standard deviation of the observation noise was set to 0.0577 for all reward and impact functions, meaning the noise was $\mathcal{N}(0, 0.0577^2)$. This is passed via the 'noise_dict' to the environment sampling step.

- **Updating GPs:** At each round $t$, after receiving a noisy observation $(\hat{R}_A, \hat{R}_B, \hat{H}_A, \hat{H}_B)$ for an allocation $a_t$, the corresponding GPs are updated by fitting them to all observations collected thus far for that function. Specifically, the 'add_point' method in 'NoisyFairnessFunction' and 'NoisyUtilityFunction' appends the new observation and re-fits the respective GP. The point $(0, 0)$ is added to the training data for $r_A, h_A$, and $(q, 0)$ for $r_B, h_B$ (by transforming $q - x_B$ to $x'_B$ and $h_B(x_B)$ to $h_B(x'_B)$ for the GP) as these are known boundary conditions.

### F.7.1 Construction of Confidence/Credible bounds

The posterior distribution from each GP, $\hat{f}(x) \sim \mathcal{N}(\mu_f(x), \sigma_f^2(x))$, is used to construct credible bounds.

- The lower bound $L_f(x) = \mu_f(x) - 1.96\sigma_f(x)$ and upper bound $U_f(x) = \mu_f(x) + 1.96\sigma_f(x)$ are used, corresponding to an approximate 95% credible interval. These bounds are then used to estimate the optimistic and pessimistic fair sets $(\hat{A}_t^+, \hat{A}_t^-)$ and the optimistic reward set $(\hat{X}_t)$ as described in the main paper. The 'find_optimistic_and_pessimistic_intervals' method in 'NoisyFairnessFunction' and 'update_optimistic_set' in 'NoisyUtilityFunction' implement this based on predictions over a discretized space (resolution=1000 points).

