# OpenReview forum: "Fair Continuous Resource Allocation with Equality of Impact"
_NeurIPS.cc/2025/Conference — NeurIPS 2025 poster_

### Official Review · Reviewer_Thu2 · 2025-06-20

**Clarity:** 3
**Significance:** 2
**Originality:** 2
**Rating:** 3
**Confidence:** 2

**Summary:**

This paper studies the following problem. There is a learning agent that receives q units of a continuous resource in every round, for T rounds. The agent distributes the resource across two groups, A and B, according to an allocation rule. Each group has a reward function and an impact function (the latter quantifies societal benefit/harm); both functions are unknown. Fairness is defined as “equality of impact,” that is, intuitively, the difference in impact between the allocation of A and B, is bounded. The goal of the agent is to maximize her utility, which is the sum of rewards.

The authors give a characterization of the optimal allocation (the interest is “no-regret” here); under assumptions (nice behavior of reward/impact + diminishing marginal returns) the authors give an algorithm with regret O(1/T) in the noise-free setting (exact values are observed after allocation). In the noisy setting, the authors give a similar algorithm, but do not formally prove guarantees.

**Questions:**

N/A

**Ethical Concerns:**

["NO or VERY MINOR ethics concerns only"]

**Final Justification:**

I appreciate the discussion with the authors. I still find overall the paper to be below the NeurIPS bar, but perhaps a score of 2 was too harsh, so I updated my score to a 3.

**Limitations:**

yes

**Paper Formatting Concerns:**

no concerns

**Quality:**

2

**Strengths And Weaknesses:**

The paper studies a natural problem. The writing is, for the most part, clear.
On the other hand, the overall contribution feels incremental. Specifically, it is not clear to me what a nice take-home message of this work is, either from a technical view (are there any new techniques or algorithmic ideas of interest?) or from a conceptual view.

---

> ### Author Rebuttal · Authors · 2025-07-31
>
> We thank the reviewer for their feedback.
>
> As summarized in the introduction, our work addresses a continuous resource allocation problem where fairness is measured by group-level outcomes rather than individual-level outcomes. This distinction is important for settings where resources (e.g., clean water) are not distributed to individuals but still affect groups unevenly. Technically, we introduce Equality of Impact as a fairness notion for this setting and propose algorithms for maximizing welfare subject to this constraint. Our setting also generalizes previous work by Elzayn et al. (2019) since we assume no prior knowledge of the welfare and impact functions (beyond their membership in a Marginal Returns class). In the non-stochastic setting, we additionally provide a sublinear fair regret guarantee on the proposed algorithm.
>
> If the reviewer has specific feedback about how to make the contributions (and take-home messages) clearer, we will be happy to incorporate it into the paper.

---

> > ### Comment · Reviewer_Thu2 · 2025-08-05
> >
> > Thank you for the response. I now better understand the intended contribution. However, I still find the overall contribution to be somewhat marginal. Introducing a new notion is primarily a conceptual contribution, not a technical one. As I understand it, the main results follow from known ideas. While the proposed notion of Equality of Impact and the relaxation of assumptions from Elzayn et al. are reasonable steps, they do not, in my view, offer enough to meet the NeurIPS bar.

---

> ### Author Response · Authors · 2025-08-08
> **Response to Reviewer**
>
> We thank the reviewer for their additional comments and will take the opportunity to clarify our contributions.
>
> * We agree that equality of impact (EoI) is a conceptual contribution, but that conceptual advances in fairness definitions are important because they can encourage future methodological contributions that address the new definition. EoI generalizes a prior fairness notion that focused on individual-level access to group-level access, which is relevant when resources aren’t directly provided to individuals but still affect groups unevenly (e.g., in environmental interventions).
> * We strongly argue that our work also includes a technical contribution—we provide fairness regret guarantees under different structural assumptions than previous work in online convex optimization or bandit literature. In particular, we show that when reward and impact functions belong to the Marginal Returns class, one can achieve a similar regret bound with the alternative information feedback (e.g. without access to full (sub)gradient information). This is distinct from prior work [1, 2, 3].
>
> We will clarify these contributions much more clearly and explicitly in the main body of the paper.
>
> [1] A low complexiy algorithm with $O(\sqrt{T})$ regret and $O(1)$ constraint violations for online convex optimization with long term constraints. Yu & Neely (2020)
>
> [2] Trading regret for efficiency: online convex optimization with long term constraints. Mahdavi, Jin, Yang (2012)
>
> [3] Adaptive algorithms for online convex optimization with long-term constraints. Jenatton, Huang, Archambeau (2016).

---

### Official Review · Reviewer_1koU · 2025-07-01

**Clarity:** 2
**Significance:** 2
**Originality:** 3
**Rating:** 4
**Confidence:** 4

**Summary:**

The paper proposes the study of fair resource allocation in an online setting when fairness is judged by the impact of allocation decisions instead of focusing on individuals. This is termed as Equality of Impact, and shown as a generalization of Allocative Equality of Opportunity. The paper derives properties of this problem setting in the presence of concave, continuous and non-decreasing utility and impact functions. The authors derive properties in the two-group case, defining Equality of Impact (EoI) in a manner similar to $\epsilon$-demographic parity. They then cast the optimization problem as a search for the most utilitarian allocation among the set of all allocations satisfying EoI, and present an algorithm to learn this allocation online during repeated allocations with minimal prior knowledge. This is studied for both the noise-free case and the noisy case, and algorithms are provided for both, showing a sub-linear regret property in the noise-free case. The paper ends with a brief empirical analysis using three environments, providing evidence of the regret bound and comparing to two baselines for the cumulative regret.

**Questions:**

Q1. How does the proposed algorithm compare to a naive baseline like line search? It seems unreasonable that the baselines have increasing cumulative regret over long horizons. They are also not compared in the non-noisy setting.

Q2. Why did the authors not consider RL or MAB-based techniques?

Q3. What would it take to extend the approach to more than 2 groups?

**Ethical Concerns:**

["NO or VERY MINOR ethics concerns only"]

**Final Justification:**

The authors addressed some of the clarity concerns and promise to update the text accordingly, and thus I will update my score to a borderline accept. I still believe that the addition of more recent and stronger baselines will be necessary to make this a strong paper.

**Limitations:**

yes

**Quality:**

3

**Strengths And Weaknesses:**

Strengths:
1. The paper explores a new kind of fairness in resource allocation with 2 groups, considering the impact of allocations instead of just the benefit. This is important and has real-world significance.
2. The theoretical properties presented characterize the problem studied in detail, which is useful for future researchers.


Weaknesses:
1. The paper is not well-written and is poorly organized. It is unclear when going through the paper why certain sections are included, and this makes it hard for readers to appreciate the intent. E.g. in section 2.1, there is no comment or exposition for why the two examples are needed. What are the authors trying to highlight?
2. There is improper or incomplete notation. For example, EoI should be a function of G (Eq.2), and the optimization objective (4) should also be stated in terms of G. It is not clear what threshold is being optimized for otherwise. Similarly, $C^i$ is not defined (line 98).
3. The paper seems to consider resources as continuous, but in some places, the exposition and examples seem to hint otherwise. E.g. in the proof for Lemma 3.1, the authors say 'all resources are allocated to group B', which suggests resources are countable and discrete, and the authors use funding (money) as an example of resources to be allocated. This may be a minor nitpick, but in this context it can matter. The authors use the Intermediate Value Theorem (IVT) to prove Lemma 3.1, and rely on the functions to be continuous, but in cases where there is discretization (like money), IVT is not valid. While it does not invalidate the paper's results, I feel it is worth pointing out.
4. The property of sublinear regret seems trivial for any learning algorithm to achieve. Even a grid-search based method will achieve sublinear regret once it lands on the optimal point, given enough time. It would be helpful to contextualize how strong the proposed algorithm's sublinearity is when compared to naive baselines like a line search.
5. Adding to the above point, the used baselines are very old. Why have the authors not used any other recent methods to solve the problem? They mention links to multi-armed bandits and RL, and these can be applied to solve the given problem.

---

> ### Author Rebuttal · Authors · 2025-07-31
>
> We thank the reviewer for their detailed (and constructive) feedback!
>
> We first respond to weaknesses 1-3, where we think the specific issues mentioned can be addressed by minor clarifications.
>
> 1. The first example in Section 2.1 (and its continuation in Appendix A) shows how our problem setting generalizes that of Elzayn et al. (2019), which we cited (starting in line 82) as motivation for our definition of Equality of Impact. The second example in Section 2.1 indicates how our problem setting could apply to the water allocation problems mentioned in the introduction (starting in line 30). We will add a sentence at the beginning of each example to make these purposes clearer. If the reviewer can identify other places where the writing or reasons for sections were unclear, we would be happy to address them in further responses.
> 1. We think the reviewer is asking us to parameterize the term "EoI", as in "$G$-EoI", since the set $A^G$ defined in (2) is already parameterized by $G$. We are happy to do this and also to mention that "EoI" without the $G$ is understood to mean EoI with a generic threshold $G$. In eq. (4), the parameter $G$ is already present in the feasible set $A^G$. In line 98, $C_i$ is the probability distribution of $c_i$, which we will clarify by changing "is a random variable" to "is a random variable with probability distribution $C_i$."
> 1. We always consider resources as continuous, as indicated even in the title of the paper, and we will try to make this even clearer. We are honestly not sure why the phrase "all resources are allocated to group B" in the proof of Lemma 3.1 led the reviewer to think that the resources might also be discrete. Regarding money as the resource, while physical money may technically be discrete (for example one cent may be the smallest unit), it is common practice to treat money as continuous and then round to the nearest unit (cent) with little loss of consequence. This is in the same spirit as footnote 1 on page 2.
>
> In the rest of this rebuttal, we respond to Q1-Q3.
>
> **Q1** (weakness 4). We assume by line search the reviewer means a naive grid or interval search over allocations that selects the highest (fair) reward. In a non-stochastic setting we agree that this baseline would converge to the optimal allocation in enough rounds. We included a better variant of this line search (the naive allocator baseline, detailed in Appendix F) that explicitly searches for allocations that satisfy fairness and then maximizes reward within that set.  Our main goal in the non-stochastic experiments was to confirm Algorithm 1’s sublinear fairness regret bound, which the naive baseline also achieves, but only after many more rounds due to exhaustive search. Our results (Figure 5) suggest that Algorithm 1 is more sample-efficient in practice (due to its leveraging of the problem structure). We will revise the paper to make this advantage over naive search methods more clear. In the noisy setting, it is unclear what a statistically principled version of line search would be (because noise can cause it to discard optimal allocations prematurely). The baselines we do compare to, while commonly used, do not have regret guarantees and may repeatedly sample sub-optimal allocations. This can lead to the accumulation of regret seen in the experimental results.
>
> **Q2** (weakness 5). The classic MAB and RL settings assume discrete actions and are therefore not directly applicable to our continuous allocation setting. While variants of these approaches, such as continuous MABs and bandit convex optimization, are more aligned with our setting, they do not address outcome-based fairness (EoI), which is an important contribution of our work. We felt that adapting these algorithms to enforce EoI would require non-trivial algorithmic changes that were more akin to designing a new method inspired by our work’s insights, as opposed to benchmarking. While we already discuss these variants in the extended related work section (Appendix H), we will additionally clarify why current methods in this space were not included in our empirical analysis.
>
> **Q3.** Some of the theoretical insights and proof techniques in this work can extend to the multi-group case. For example, if all reward and impact functions satisfy Assumption 2.2, then the fairness constraint may continue to define a closed and convex feasible region. Moreover, because the overall reward remains a sum of concave functions (subject to linear resource constraints), the optimization generalizes to a concave maximization over a convex set. This implies that Theorem 3.3 may generalize to a $k > 2$ group setting. We believe the main challenge is scaling the estimation of the (guaranteed and potential) fairness sets,  as well as the utility-maximizing set, in higher dimensions. Developing online methods for these estimations is an important future direction.
>
> We thank the reviewer again for their feedback and hope our clarifications and planned revisions address the concerns raised.

---

> > ### Comment · Reviewer_1koU · 2025-08-05
> >
> > I thank the authors for their detailed feedback. I appreciate some of the proposed changes, and agree that they will add strength to this paper. I think adding details of a naive grid search to the main text, and also contextualizing the analysis based on this will be useful as well. (e.g. how much better than a naive strategy is it, since both have sublinear regret, in the non-stochastic case).
> >
> > I will update my score to reflect this rebuttal.

---

> > > ### Author Response · Authors · 2025-08-08
> > >
> > > We thank the reviewer for updating their score in light of our planned changes. As a comment on their last message—we agree that adding discussion of the sublinear regret of the naive method in the main body is important. The additional page allowance for accepted papers would also make it easier to move experimental details on the naive algorithm from the appendix into the main text, so that its performance and comparison to our method are clearer.

---

### Official Review · Reviewer_VNt7 · 2025-07-02

**Clarity:** 3
**Significance:** 2
**Originality:** 2
**Rating:** 4
**Confidence:** 4

**Summary:**

On an abstract level, the authors consider an online constrained optimization problem. There are two groups where each group has a reward function and an impact function. Initially these functions are unknown. Then in each round, the agent selects an action (resource allocation) for both groups subject to a capacity constraint. The objective is to maximize the sum of the rewards. There is an additional “equality of impact” constraint: requiring that for each round, the difference in impact of the two groups is no larger than a constant G. To evaluate the effectiveness of an algorithm, the authors consider both the regret in reward compared to the optimal feasible allocation, and the fairness regret, which is the total violation in the equality constraint.

Under concavity assumptions, the authors show that the optimal action for each period should use all resources, turning the original two-dimensional decision into one-dimensional. Using this property and the concavity assumptions, this paper constructs confidence bounds for the reward and impact functions, using which it gives an algorithm with constant fairness regret. It then discusses heuristics to extend the algorithm to a noisy setting and offers several numerical results.

**Questions:**

1.	Why the existing approaches from online convex optimization with constraints do not work for your setting?

2.	To what extent the proposed algorithm may extend to multiple groups?

3.	What guarantee for the regret on reward does the proposed algorithm have? Is it still a constant independent of the time horizon like the fairness regret result?

4.	Would Assumption 6.1 help give regret bound for the noisy setting?

**Ethical Concerns:**

["NO or VERY MINOR ethics concerns only"]

**Final Justification:**

This paper has nice contribution to the literature of fair allocation. The authors' response addresses my concern on the relationship with the literature of online convex optimization. However, I still have concerns about the completness of the results, especially the lack of guarantees on reward. Given such concerns, I am keeping my scores at 4.

**Limitations:**

yes

**Quality:**

3

**Strengths And Weaknesses:**

Strength:

1.	The model in this paper generalizes previously studied fair allocation problems such as allocative equality of opportunity and water allocation.

2.	The way to construct confidence bounds is pretty intuitive and the algorithm appears to be efficient in shrinking the set of optimal solutions.

3.	Numerical results support the benefit of the proposed algorithm.

Weakness:

1.	The studied model seems to be a special case of online convex optimization with long-term constraints. It is typical that one can achieve square-root T regret guarantee and constant fairness regret (constraint violation); see for example, https://www.jmlr.org/papers/v21/16-494.html. The authors should better compare their results with this literature and be clear why this work presents new results.

2.	Moreover, the model is rather restrictive in the sense that only two groups are considered. It is not explained why the setting of two groups is sufficiently interesting and captures real-world applications (such as the listed application of water allocation).

3.	In addition, although the paper bounds the fairness regret in Theorem 5.2, there is no guarantee on the regret. There is also no guarantee for the noisy setting.

---

> ### Author Rebuttal · Authors · 2025-07-31
>
> We thank the reviewer for their feedback and have addressed the mentioned weaknesses and questions in our rebuttal. We are happy to provide further clarification or continue the discussion if any concerns remain.
>
> **Q1.** We thank the reviewer for directing us to work in online convex optimization (OCO) with long-term constraints. While we mention OCO in the extended related work (appendix H), we agree that explicitly including the provided example [1] and similar related works [2, 3], as well as clarifying how our contributions differ, is important. Our problem can be viewed as a constrained OCO problem: maximizing utilitarian welfare corresponds to minimizing a convex loss function, and our fairness constraint (EoI) can be formulated as functional inequality constraints.
>
> Importantly, [1-3] make informational assumptions that do not hold in our setting. For example, [1] assumes access to explicit (sub)gradients of the objective and constraint functions after each allocation decision (see Assumption 1 and footnote 1 in [1]), which allow for standard primal-dual gradient updates. This is not assumed in our problem setting. Instead, we observe only point-wise function outcomes, and leverage the properties of the Marginal Returns class to construct upper and lower bounds from these limited samples. This allows us to estimate the feasible sets online and without access to (sub)gradient oracles.
>
> Our approach also differs from [1-3] with respect to fairness regret guarantees. The way we exploit the structure of the Marginal Returns class allows us to achieve the $O(1/T)$ cumulative fairness regret guarantee, which implies a faster asymptotic (average) decay than the rates obtained in [1-3]. In conclusion, while our problem can be framed as an instance of OCO, our results show that by considering EoI under an alternative information feedback model than those previously considered in OCO literature, we can obtain tighter fairness guarantees. We plan to make these these distinctions and contributions much more clear in the main paper.
>
> [1] A low complexiy algorithm with $O(\sqrt{T})$ regret and $O(1)$ constraint violations for online convex optimization with long term constraints. Yu & Neely (2020)
>
> [2] Trading regret for efficiency: online convex optimization with long term constraints. Mahdavi, Jin, Yang (2012)
>
> [3] Adaptive algorithms for online convex optimization with long-term constraints. Jenatton, Huang, Archambeau (2016).
>
> **Q2.** We recognize that this setting limits the real-world scope, though it can be relevant in domains where inequity between two main populations is a concern (e.g., disadvantaged vs majority groups). Our goal in this work is to lay a theoretical groundwork for learning fair allocations (specifically, allocations that satisfy EoI) under partial feedback. This is important when resources (such as clean water) are not distributed directly to individuals but can still unevenly affect groups. Some of the theoretical insights and proof techniques in this work can extend to the multi-group case: If all reward and impact functions satisfy Assumption 2.2, then the fairness constraint would still define a closed and convex feasible region. Moreover, because the overall reward remains a sum of concave functions (subject to linear resource constraints), the optimization generalizes to a concave maximization over a convex set. This implies that Theorem 3.3 may generalize to a $k > 2$ group setting. The main challenge is scaling the estimation of the (guaranteed and potential) fairness sets, as well as the utility-maximizing set, in higher dimensions. This remains an important future direction, and it is possible that insights from [1] could be useful.
>
> **Q3.** We thank the reviewer for this question and agree that clarifying the guarantees on reward regret is important. In this work, our primary focus is on providing a strong fairness violation guarantees (the $O(1/T)$ cumulative bound for EoI violations). This result holds under a small set of structural assumptions on the reward functions (the Marginal Returns Class properties).
>
> Our algorithm does nt explicitly provide reward regret rates. In online convex optimization with long-term constraints, sublinear reward regret typically requires additional assumptions such as smoothness, strong convexity, or bounded (sub)gradients (e.g., [1, 2]). If our setting were to satisfy these additional conditions, e.g., if the reward functions were strongly convex, our framework could potentially be extended to also provide sublinear reward regret guarantees. However, this is outside the immediate scope (and main contribution) of our current work. We appreciate this suggestion and will expand the appendix to discuss how introducing further structural assumptions could lead to reward regret guarantees.
>
> **Q4.** We found Assumption 6.1 useful for our empirical analysis but did not explore its use in deriving regret bounds for the noisy setting. We agree that it could potentially help in this direction, but believe additional assumptions on the estimation procedure would be required. For example, in the Gaussian process setting considered in this work, additional assumptions on the kernel choice and noise properties would likely be necessary to establish regret bounds. Providing these guarantees is an important future direction.

---

> > ### Comment · Reviewer_VNt7 · 2025-08-01
> >
> > I want to thank the authors for their detailed response. However, as far as I understand, the cumulative fairness violation guarantee in this paper is O(1), not O(1/T), right? (Theorem 5.2 in Appendix D.1) In this case, this guarantee is not stronger than [1], who also proved constant constraint violation. I do believe that this paper requires less assumption than prior work, but it is unclear to me if the results are as complete as the authors claim. For example, I view the missing of reward regret guarantee as a major shortcoming of this paper.
> >
> > [1] A low complexiy algorithm with $O(\sqrt{T})$ regret and $O(1)$ constraint violations for online convex optimization with long term constraints. Yu & Neely (2020)

---

> > > ### Author Response · Authors · 2025-08-06
> > > **Response to Reviewer**
> > >
> > > We thank the reviewer for this correction and sincerely apologize for the error in our response. The reviewer is correct: our algorithm’s guarantee of average fairness regret of $O(1/T)$ (which corresponds to a cumulative fairness regret) is in fact comparable to the $O(1)$ constraint violation provided in [1]. We still believe our work remains an important contribution to the literature because it addresses a relevant and distinct setting from prior OCO work. Specifically:
> > >
> > > * We introduce equality of impact with the goal of generalizing a resource allocation fairness metric (an equality of opportunity variant) to focus on group-level outcomes rather than individual-level outcomes. We feel this perspective is important because some critical resources that are not directly provided to individuals can nonetheless be unevenly available to different groups.
> > > * Our work makes alternative assumptions to prior work. Instead of assuming gradient information of the objective and constraint functions, our method assumes that reward and impact functions belong to the Marginal Returns class. This allows us to create reliable bounds on relevant functions from different feedback information, and ultimately derive similar constraint regret guarantees to [1].
> > >
> > > As stated previously, we plan to clarify the relationship between our work and existing work in OCO literature more thoroughly than what is currently in Appendix H, and include this information in the main body of the paper.
> > >
> > > Lastly, we understand the reviewer’s concern regarding the lack of reward regret guarantee. Beyond what we stated in our original response—that in prior work, sublinear reward regret typically requires additional assumptions such as smoothness, strong convexity, or bounded (sub)gradients, and that we plan to discuss these extensions—we agree that providing a formal theoretical guarantee for reward regret is important, though it falls outside the primary focus of the present work. We view this as an important future direction.
> > >
> > > Again we thank the reviewer for their response and hope we have resolved any confusion caused by the earlier error.

---

### Official Review · Reviewer_bxt8 · 2025-07-13

**Clarity:** 4
**Significance:** 3
**Originality:** 3
**Rating:** 5
**Confidence:** 3

**Summary:**

This paper addresses fair continuous resource allocation in settings where fairness should be measured by comparing the impact of allocations on groups rather than by comparing the impact of allocations on individuals between groups. This is relevant because some important resources, such as clean water or air, are not allocated directly to individuals but are nevertheless inequitably available to different groups.  The authors introduce Equality of Impact, a generalization of Equality of Opportunity, as a measure of the fairness property they introduce and develop algorithms for both noise-free and noisy settings that achieve sublinear regret in optimizing for Equality of Impact.

**Questions:**

SCALE:
(a) It would be helpful to say more about generalization to the >2 group setting. Many real world settings have more than two relevant groups and have potentially complex group structures, such as overlapping groups or subgroups formed by intersectional identities.  While the paper need not address all these cases, it would be helpful to have pointers to how the results are expected to generalize.

(b) I also wonder how the exhaustive search method scales to higher dimensional problems or more complex fairness complaints. Mentioning this aspect of scale might also be helpful in determining the real-world utility of the techniques.

EXPERIMENTS: While the experiments are described primarily in the appendix, they provide important support for the claims of the paper. If I am understanding correctly that both experiments are on synthetic data, I am curious about this choice and would like to understand why neither used empirical real-world data. In my view, using real data would strengthen the evidentiary value of the experimental results.

FAIRNESS TRADEOFFs: It would be great to have more detail about the tradeoff between fairness and welfare maximization. In particular, how sensitive is the welfare maximization performance to the choice of the fairness metric?  Do some fairness metrics force greater welfare losses than others, or does the severity of the tradeoff depend on the underlying structure of the data?

**Ethical Concerns:**

["NO or VERY MINOR ethics concerns only"]

**Final Justification:**

The other reviewers raised concerns about the scope and novelty of the paper's formal contribution, so I will not raise my score, but I did find the author responses helpful and feel that they addressed my questions.

**Limitations:**

Yes.

**Paper Formatting Concerns:**

No major formatting issues

**Quality:**

3

**Strengths And Weaknesses:**

STRENGTHS: The paper introduces a fairness characterization and metric, which is in itself an original contribution. The paper motivates why this metric is needed in a compelling way and provides a rigorous characterization.  The main theoretical result for the non-noisy setting algorithm (Algorithm 1), the demonstration that the algorithm provides a sublinear regret rate and therefore approaches optimality in the long term, is clearly described.

WEAKNESSES: The scope of applicability and scalability to more complex scenarios is not yet explored, leaving it somewhat unclear how useful the metric could be in a real-world scenario. The fact that the experiments appear to use synthetic data (or are so briefly described that it is hard to tell what kind of data they use) does not help with the projection to potential use cases. I think both of these concerns can be addressed, however.

---

> ### Author Rebuttal · Authors · 2025-07-31
>
> We appreciate the reviewer’s feedback, and focus on answering their questions below.
>
> **SCALE.** We recognize that the two-group setting limits the real-world scope, though it can be relevant in domains where inequity between two main populations is a concern (e.g., disadvantaged vs majority groups). Our goal in this work is to lay a theoretical groundwork for learning fair allocations (specifically, allocations that satisfy EoI) under partial information feedback. Some of the theoretical insights and proof techniques in this work can extend to the multi-group case: If all reward and impact functions satisfy Assumption 2.2, then the fairness constraint would still define a closed and convex feasible region. Moreover, because the overall reward remains a sum of concave functions (subject to linear resource constraints), the optimization generalizes to a concave maximization over a convex set. This implies that Theorem 3.3 may generalize to a $k > 2$ group setting. The main challenge is scaling the estimation of the (guaranteed and potential) fairness sets, as well as the utility-maximizing set, in higher dimensions. This remains an important future direction.
>
> We will also expand discussion of related work to include literature on different group dynamics, including overlapping and intersectional groups. While these extensions are beyond the scope of the current work, we believe it can be used as a stepping stone to more complex group dynamics.
>
> Regarding the “exhaustive search method,” we assume the reviewer is referring to line 9 of Algorithm 2. This step does not require enumerating all allocations---in the two-group setting, the maximizers occur at intersection points of neighboring allocations (or at the allocations themselves), which can be found algebraically. We will clarify this point in the paper. In higher-dimensional settings this search step will be more computationally demanding.
>
> **EXPERIMENTS.** We agree that using real-world data would strengthen this work. Our choice to use synthetic data was driven by the lack of publicly available datasets that include group- (or relevant individual)-level outcome functions for our setting. We considered, for example, the Philadelphia crime dataset used by Elzayn et al. (2019), which considers a related fairness constraint. However, synthetic data was used to model the relationship between outcomes and allocations. We followed this precedent and designed synthetic experiments to capture realistic patterns reported in prior studies (see Section 7 for more information). We will further clarify this reasoning in the paper.
>
> **FAIRNESS TRADEOFFS.** Yes, as the reviewer notes, the tradeoff arises because imposing EoI can restrict allocations away from welfare-maximizing solutions (i.e., when the sets $\mathcal Q^G$ and $X$ do not overlap), and the severity of this depends on the behavior of the underlying impact and reward functions. It also depends on $G$, which controls the allowable impact disparity. Stricter fairness requirements will tend to shrink $\mathcal Q^G$ which may result in a larger sacrifice in welfare. While the role of $G$ is briefly described in lines 94-95, we will further elaborate on this tradeoff in the appendix. If the reviewer is referring to a different aspect of the fairness-welfare tradeoffs, we would be happy to address this as well.

---

> > ### Comment · Reviewer_bxt8 · 2025-08-08
> >
> > Thank you -- I appreciate the authors commitment to discussing different group dynamics, including overlapping and intersectional groups, and to clarifying the methodological choices involved including the choice to use synthetic data. I also found the responses to other similar comments, like VNt7's Q2 about the choice to use two groups, helpful. Although I haven't changed my score, I think the changes described would strengthen the paper.

---

> > > ### Author Response · Authors · 2025-08-08
> > >
> > > We appreciate the reviewer’s positive assessment and thank them for their insights and suggested improvements. We agree that these changes will strengthen our work, and are happy to include them.

---

### Decision · Program_Chairs · 2025-09-17

**Decision:**

Accept (poster)

**Comment:**

The paper addresses the important and timely problem of fair resource allocation at the group level, introducing "Equality of Impact" (EoI) as a fairness measure and analyzing algorithms that optimize allocations under this notion. Reviewers both acknowledge the conceptual value of shifting fairness from individual to group outcomes and commend the clean theoretical guarantees under weaker feedback assumptions than much of the prior constrained optimization literature. However, the consensus emerges that the technical novelty relative to prior work is modest: key regret bounds align with known results, and the main distinction is informational rather than algorithmic. The paper's experimental section is based entirely on synthetic data, limiting its empirical strength, and generalization to multi-group or intersectional settings remains unaddressed. There is also room for improvement in clarity, organization, and inclusion of stronger contemporary baselines. Despite these limitations, the authors have provided clarifying rebuttals and promised significant revisions. The paper is recommended for acceptance, primarily on the value of the conceptual contribution and the potential for new lines of fairness research. However, the authors are strongly recommended to pay attention to the planned clarifications and empirical improvements for the camera ready, should include a few new baselines, clarify the comparison to prior online convex optimization work, and explain how the approach may (or may not) extend to more complex group structures.